# NoMod: A Non-modular Attack on Module Learning With Errors

## Abstract

The advent of quantum computing threatens classical public-key cryptography, motivating NIST's adoption of post-quantum schemes such as those based on the Module Learning With Errors (Module-LWE) problem. We present NoMod ML-Attack, a hybrid white-box cryptanalytic method that circumvents the challenge of modeling modular reduction by treating wrap-arounds as statistical corruption and casting secret recovery as robust linear estimation. Our approach combines optimized lattice preprocessing—including reduced-vector saving and algebraic amplification—with robust estimators trained via Tukey's Biweight loss. Experiments show NoMod achieves full recovery of binary secrets for dimension $n = 350$, recovery of sparse binomial secrets for $n = 256$, and successful recovery of sparse secrets in CRYSTALS-Kyber settings with parameters $(n, k) = (128, 3)$ and $(256, 2)$. We release our implementation in an anonymous repository https://anonymous.4open.science/r/NoMod-3BD4.

## 1 Introduction

The dawn of quantum computing presents a significant and growing threat to current cryptographic systems, many of which may be vulnerable to decryption through quantum-based attacks. At the heart of this risk is Shor's algorithm, a quantum-based algorithm developed in 1994 by Peter Shor, which can efficiently factor large integers and compute discrete logarithms. These two mathematical problems are computationally challenging for classical computers when the input size is large. In particular, while classical algorithms to factor integers, such as the General Number Field Sieve (GNFS), run in sub-exponential time, Shor's algorithm could run in polynomial time, when implemented on a sufficiently robust quantum computer Shor (1994; 1997).

This development poses a significant threat to the security assumptions underlying widely used public-key cryptographic schemes, such as RSA, Elliptic Curve Cryptography (ECC), and the Diffie-Hellman key exchange. These algorithms are central to the Public Key Infrastructure (PKI) that secures virtually all modern digital communications. Although no practical quantum computers currently exist with the required number of qubits and sufficiently low error rates to implement Shor's algorithm at the scale needed to compromise modern public-key schemes, ongoing improvements in quantum hardware, fault-tolerant architectures, and algorithmic optimization indicate that this capability may arise within the following decades Campagna et al. (2021). In response to this emerging challenge, researchers, cryptographers, and standardization bodies have mobilized to design, evaluate, and implement cryptographic algorithms that can resist quantum attacks. This new generation of cryptography, post-quantum cryptography (PQC), embraces cryptographic primitives that can be securely installed on classical hardware while being resilient against quantum adversaries. The National Institute of Standards and Technology (NIST) launched the Post-Quantum Cryptography Standardization Process to identify public-key algorithms resistant to adversaries equipped with large-scale quantum computers. This multi-year effort combines broad public participation, sustained cryptanalysis, and NIST's own evaluation of security, efficiency, and practical implementability.

In 2022, NIST announced its first algorithm selections: the Module-Learning With Errors (Module-LWE or MLWE) Key Encapsulation Mechanism (KEM) CRYSTALS-Kyber, the Module-LWE signature scheme CRYSTALS-Dilithium, and the hash-based signature scheme SPHINCS+.[1] Given

---

[1]On March 11, 2025, NIST selected Hamming Quasi-Cyclic (HQC) for standardization as an additional KEM, adding a code-based primitive to the PQC suite.

that two of the four standardized post-quantum algorithms are built on the Module-LWE problem, understanding and continually reevaluating the security of this family is critical. The hardness of MLWE has become a foundation of post-quantum security, with classical and quantum attacks significantly studied and security margins broadly established under current models. However, the rapid progress of artificial intelligence (AI) has introduced a new dimension: AI-powered approaches to cryptanalysis, still in their infancy, have shown an extraordinary ability to uncover patterns that resist traditional analysis, which raises a compelling question: *could machine learning methods reduce the hardness of MLWE, either by directly learning modular correlations or by exploiting alternative representations that uncover hidden structures?*

A central technical barrier in applying machine learning to the MLWE problem lies in the modular arithmetic inherent to its structure. In contrast to classical regression or signal recovery tasks, where adversaries could directly exploit linear relationships between variables, the reduction modulo $q$ disrupts linearity and introduces non-linear wrap-around effects that are difficult for neural models to capture. This work advances the study of Module-LWE security under machine learning-based attacks by combining lattice reduction techniques with robust statistical learning. Our main contributions are:

1. We introduce a novel "NoMod" approach that avoids modular arithmetic by treating wrap-around effects as statistical outliers. By re-framing the problem into a noisy, yet linear domain, this strategy enables efficient secret recovery through lattice reduction combined with regression, offering a light alternative to black-box transformer-based attacks.
2. We perform a systematic study of the preprocessing pipeline, identifying key trade-offs between reduction quality, sample size, and computational cost. Several optimizations are proposed, including progressive Block Korkine–Zolotarev (BKZ) strategies with moderate block sizes, analytical determination of the optimal sample count per reduced matrix, and accumulation of short vectors across multiple reduction tours.
3. We exploit the automorphism structure of polynomial rings to amplify a small set of reduced samples into a full orbit, lowering the number of lattice reductions required. This technique highlights how algebraic properties of the problem can be directly harnessed to strengthen machine learning–based attacks.
4. Our analysis demonstrates that robust regression can be tuned to reliably extract secrets from noisy reductions, even in the presence of modular wrap-around effects. It establishes robust linear methods as a foundation for AI-powered cryptanalysis, contrasting with black-box and computationally expensive transformer-based models.

## 2 PRELIMINARIES

### 2.1 LEARNING WITH ERRORS

The LWE problem, firstly introduced by Regev in 2005 Regev (2005; 2010), is a central hardness assumption in lattice-based cryptography and serves as the foundation for many post-quantum secure cryptographic primitives. Informally, LWE can be seen as the problem of solving a noisy system of linear equations over a finite field. Without noise, such systems are easily solvable in polynomial time using standard linear algebra. The introduction of slight random noise makes the problem computationally difficult under appropriate parameter choices, even for quantum adversaries.

**Search-LWE.** Let $\mathbb{Z}_q = \{0, 1, \ldots, q-1\}$ denote the ring of integers modulo $q$. Fix positive integers $n$ and $m$, an integer modulus $q \geq 2$, and an error bound $B \ll q/2$. Let:

- $s \xleftarrow{\$} \mathbb{Z}_q^n$ be a *secret vector* chosen uniformly at random,
- $A \xleftarrow{\$} \mathbb{Z}_q^{m \times n}$ be a uniformly random matrix,
- $e \xleftarrow{\$} [-B, B]^m$ be an *error vector* whose entries are small integers.

We compute $b = As + e \pmod{q}$. The problem *search-LWE*, denoted $\text{LWE}(m, n, q, B)$, is that given $(A, b)$, one must recover the secret vector $s$. The parameters must be chosen carefully. If $B = 0$ (i.e., $e = 0$), then solving LWE reduces to solving a system of modular linear equations. Conversely, if $B \geq (q-1)/2$, the error term completely masks the signal, making recovery of the $s$ information theoretically impossible. In cryptographic applications, $B$ is chosen so that $B \ll q/2$

and $m \gg n$ to ensure uniqueness of the solution with overwhelming probability Regev (2010). Short secrets may be drawn from other constrained distributions in practice to improve performance and reduce key sizes: 1) binary-secret LWE: $s \in \{0, 1\}^n$, 2) ternary-secret LWE: $s \in \{-1, 0, 1\}^n$, and 3) CBD-secret LWE: $s$ is sampled from a *centered binomial distribution* (CBD), producing small, approximately Gaussian-like coefficients. These variants often preserve LWE's assumed hardness for appropriate parameters and are widely used in lattice-based cryptosystems Brakerski et al. (2013).

## 2.2 LWE STRUCTURAL VARIANTS

The *Ring-Learning With Errors* (Ring-LWE) problem generalizes the classical LWE problem from vectors over $\mathbb{Z}_q$ to elements in a polynomial ring modulo $q$. This implies more compact key sizes and faster computations due to the algebraic structure of the ring Lyubashevsky et al. (2010).

Let $n$ be a power of two and let $R_q = \mathbb{Z}_q[x]/(x^n + 1)$ be the $n$-th cyclotomic integer ring. Fix an error bound $B \ll q/2$ and a probability distribution $\chi$ supported on *small* polynomials in $R_q$. Let:

- $s \xleftarrow{\$} R_q$ be a *secret polynomial* chosen uniformly at random,
- $a \xleftarrow{\$} R_q$ be a uniformly random *public polynomial*,
- $e \xleftarrow{\$} \chi$ be an *error polynomial* with small coefficients drawn from $\chi$,

we define $b = a \cdot s + e \in R_q$, where the operations $\cdot, +$ denote the product and sum in $R_q$. The *search Ring-LWE* problem is that given samples $(a, b) \in R_q \times R_q$, one must recover $s$.

The *Module-Learning With Errors* (Module-LWE) problem generalizes both LWE and Ring-LWE by working over modules of rank $\ell$ over the polynomial ring $R_q$. It can be seen as replacing the polynomials in Ring-LWE with *vectors* of polynomials in $R_q$, reducing the algebraic structure compared to Ring-LWE, allowing flexible trade-offs between efficiency and security Brakerski et al. (2011).

Let $n$ be a power of two, $R_q = \mathbb{Z}_q[x]/(x^n + 1)$, and $\ell, k \in \mathbb{N}$ with $k \geq \ell$. Fix an error bound $B \ll q/2$. Let:

- $s \xleftarrow{\$} R_q^\ell$ be a *secret vector* of $\ell$ polynomials chosen uniformly at random,
- $a_1, a_2, \ldots, a_k \xleftarrow{\$} R_q^\ell$ be $k$ uniformly random *public vectors* of $\ell$ polynomials,
- $e \xleftarrow{\$} S_B^k$ be an *error vector* of $k$ polynomials whose coefficients lie in $[-B, B] \subset \mathbb{Z}_q$.

For each $i \in \{1, \ldots, k\}$, compute $b_i = a_i^T s + e_i \in R_q$. The *search-Module-LWE problem* is, given $(a_1, \ldots, a_k, b_1, \ldots, b_k)$, one must to recover $s$. When $k = \ell = 1$, the Module-LWE problem reduces exactly to a single instance of the Ring-LWE problem.

## 2.3 LATTICES

Lattices are discrete subgroups of $\mathbb{R}^n$ with rich algebraic and geometric structure, and they form the mathematical foundation underlying the hardness of LWE. Let $B = \{v_1, v_2, \ldots, v_m\} \subset \mathbb{R}^n$ be a set of $m \leq n$ linearly independent vectors. The *lattice* generated by $B$ is:

$$L(B) = \left\{ \sum_{i=1}^m x_i v_i \ : \ x_i \in \mathbb{Z} \right\}.$$

The set $B$ is called a *basis* of $L(B)$, the rank of $L(B)$ is $m$, and if $m = n$ the lattice is called *full-rank*. The *volume* of a lattice $L(B)$, also called the *lattice determinant*, is defined as $\mathrm{vol}(L) = \sqrt{\det(B^\mathsf{T} B)}$. If $L$ is *full-rank*, this simplifies to $\mathrm{vol}(L) = |\det(B)|$. The volume is an invariant of the lattice: it does not depend on the choice of basis. Intuitively, it measures the "density" of the lattice points in $\mathbb{R}^n$: a larger volume corresponds to a sparser lattice, while a smaller volume indicates that the lattice points are more densely packed. Beyond their geometric interest, many computational problems on lattices are central to the study of their algorithmic complexity and play a role in the foundations of lattice-based cryptography Micciancio & Goldwasser (2002).

**Unique Shortest Vector Problem (uSVP)**. Let $L \subset \mathbb{R}^n$ be a lattice of rank $n$ given by a basis $B$. For a parameter $\gamma > 1$, the $\gamma$-*uSVP* asks, given $B$, to find a shortest nonzero vector $v \in L$ under the

promise that $\lambda_2(L) \geq \gamma \cdot \lambda_1(L)$, where $\lambda_1(L)$ and $\lambda_2(L)$ are the first and second successive minima of $L$, respectively. That is, the shortest nonzero vector is unique up to sign and is at least a factor $\gamma$ shorter than all other linearly independent vectors in $L$.

A lattice reduction algorithm transforms a basis $B = \{b_1, \ldots, b_n\}$ of a lattice $\mathcal{L} \subset \mathbb{R}^n$ into a basis $B' = \{b'_1, \ldots, b'_n\}$ of relatively short and nearly orthogonal vectors. The goal is not necessarily to find the shortest vector, which is computationally hard, but to transform the basis into a form that is easier to work with. For more information about lattice reduction techniques, see Appendix A.

### 2.4 CRYSTALS-KYBER

CRYSTALS-Kyber is a quantum-safe Key Encapsulation Mechanism, standardized by NIST in FIPS 203 National Institute of Standards and Technology (2024) under the name ML-KEM (Module-Lattice-based KEM), because it is based on the hardness of the MLWE problem. The Kyber KEM is derived from the Kyber Public Key Encryption (Kyber-PKE) scheme through the Fujisaki–Okamoto transform, achieving chosen-ciphertext security from the underlying chosen-plaintext secure encryption. Kyber comes in three standardized parameter sets, corresponding to different security categories (NIST Levels 1, 3, and 5) Avanzi et al. (2021). Each set specifies the dimension parameter $k$, the noise sampling parameters $\eta_1$ and $\eta_2$ for the secret and error polynomials during key generation and key encapsulation, and the compression parameters $d_u$ and $d_v$ used for the two components of the ciphertext. These parameters balance security, bandwidth, and performance, with larger $k$ values yielding higher security levels at the cost of increased computational and memory requirements. In our attack, we will target the key generation process, particularly the recovery of $\mathbf{s}$ from $(A, \mathbf{b})$, since this directly undermines the decapsulation procedure of Kyber-KEM.

### 2.5 ROBUST ESTIMATORS

In this work, we focus on linear and robust linear models for secret recovery rather than transformer-based architectures employed in previous machine learning-based attacks Wenger et al. (2022); Li et al. (2023b;a); Stevens et al. (2024). While transformer models can approximate modular arithmetic operations, they act as black-boxes: the learned weights do not directly correspond to secret components, and secret extraction requires both large amounts of training data and additional post-processing. In contrast, linear models are inherently *white-box*. Once trained, their learned coefficients directly encode the secret vector $\mathbf{s}$, allowing immediate recovery without auxiliary algorithms. This transparency significantly reduces both computational and memory requirements, enabling fast interleaved recovery during preprocessing. Moreover, linear regression is constructed to capture the underlying linear relationship $\mathbf{b} = A\mathbf{s} + \text{noise}$, and robust variants allow us to treat modular wrap-around as statistical outliers. We discuss several regressor techniques in Appendix B.

## 3 METHODOLOGY

### 3.1 PREPROCESSING

We first transform the RLWE and MLWE instances into blocks of standard LWE samples using the transformations described in Appendices C and D. Before attempting sample recovery, we apply a preprocessing stage based on lattice reduction. The goal is to transform the LWE instance into one with smaller coefficient magnitudes and thereby reduce the effective variance of the unreduced right-hand side $b$. Since the success of the unwrapping step depends critically on the distribution of the transformed $A$ matrix, this preprocessing is essential to bring the samples into a regime where likelihood-based recovery becomes feasible. We embed the LWE matrix $A \in \mathbb{Z}_q^{n \times n}$ into a higher-dimensional lattice basis, apply lattice reduction, and obtain a unimodular transformation matrix $[R \quad C]$. This matrix yields a new instance $(RA, Rb)$ with transformed error $Re$, whose variance depends directly on the quality of the reduction. Different embeddings of $A$ govern how effectively reduction can shrink the norms of the rows of $RA + qC$, and therefore directly affect the trade-off between error amplification and variance reduction.

**Parameters**. As in previous works Li et al. (2023b;a); Stevens et al. (2024), we preprocess the LWE instances via an error-penalized dual embedding: for each sampled matrix $A \in \mathbb{Z}_q^{m \times n}$, we construct

$$\Lambda = \begin{bmatrix} \omega \cdot I_m & A \\ 0 & q \cdot I_n \end{bmatrix},$$

where the penalty parameter $\omega$ regulates the trade-off between the reduction strength on $A$ and error amplification in the transformed instance. After BKZ 2.0 reduction, the basis takes the form $\begin{bmatrix} \omega R & RA + qC \end{bmatrix}$, yielding new LWE samples $(RA, Rb)$ with error vector $Re$. Empirically, moderate values of $\omega$ maximize decoding power: small $\omega$ favors short vectors but induces excessive error growth, while large $\omega$ suppresses noise at the cost of weaker reduction; in practice, $\omega = 4$ suffices for CBD errors, whereas Gaussian errors with $\sigma = 3$ require $\omega = 10$. Reduction is performed in two phases to ensure both stability and efficiency. We first apply four iterations of FLATTER with low compression ($\alpha = 0.001$), which incrementally improves orthogonality without destroying structural correlations, then switch to BKZ 2.0 with $\delta = 0.99$ and a progressive block-size schedule from 20 to 40. Between BKZ tours, we apply the polish routine, which consistently sharpens the basis without undoing progress. Block sizes are adapted conservatively: when progress stalls for four tours, we increase the block size by increments of 10, thereby maintaining steady improvement without incurring prohibitive runtime per tour. Finally, the optimal number of samples $m$ is determined by minimizing the Gaussian-heuristic estimate of the shortest vector length, leading to the closed-form guide $m = \sqrt{\frac{n \cdot k \cdot (\log q - \log \omega)}{\log \gamma_0}} - n \cdot k$, where $\gamma_0$ captures the root-Hermite factor achieved by BKZ. This expression highlights the central trade-off: larger modulus-to-noise ratios allow more samples to be exploited, while weaker reduction quality forces $m$ downward.

**Vector Saving Strategy**. To further reduce the average norm of the output vectors beyond what a standard BKZ 2.0 schedule can offer, we implement a modified reduction pipeline that retains and accumulates short vectors across multiple tours. Unlike standard lattice reduction on LWE, which overwrites the working basis at each tour and discards previously discovered vectors, our strategy selectively retains the shortest unique vectors seen throughout the entire reduction process. This approach shifts the focus from producing a fully reduced basis to generating a high-quality set of short vectors, suitable for statistical inference in our machine learning pipeline.

Concretely, after each BKZ 2.0 tour, we extract candidate short vectors from the current basis and evaluate them based on the resulting approximate $\sigma_{\tilde{b}}$, which serves as the priority metric. A bounded priority queue maintains the best vectors seen so far, with a fixed capacity to limit memory usage and computational overhead. In particular, the queue saves only the vectors with a priority lower than the current maximum in the queue, while it also checks and discards duplicates. Over multiple tours, this strategy produces a collection of diverse short vectors with significantly reduced average norm compared to any single tour of standard BKZ 2.0. This process brings the distribution of the retained vectors closer to the theoretical shortest vector length of BKZ. While this strategy sacrifices the output being an orthogonal basis of the original embedded lattice (since many of the saved vectors are not mutually reduced or necessarily orthogonal), it aligns well with our application goal. Our attack does not require a basis, but only a collection of vectors with small norm and correct structure, allowing us to extract approximate LWE samples in the clear (i.e., without modular reduction). In this context, the breakdown of basis structure is a worthwhile trade-off for obtaining a tighter distribution on $R\tilde{b}$, which in turn boosts the effectiveness of our downstream machine learning attack.

## 3.2 ENHANCING MLWE

We enhance the Module-LWE attack by resampling rows to increase diversity, projecting $q$-ary matrices to remove dependencies, and exploiting negative-circulant structure to generate additional short vectors.

**Resampling Method: Polynomial-Row Subsamples, Offsets, and Coverage**. We treat each MLWE sample $(\mathbf{a}, b)$ with $\mathbf{a} = (a^{(1)}, \ldots, a^{(k)}) \in R_q^k$ in its coefficient embedding $\iota : R_q \hookrightarrow \mathbb{Z}_q^n$ and view every polynomial row $a^{(i)}$ as an independent source of $n$ coefficient-rows. Let the available coefficient space be partitioned into $B$ circulant blocks of length $n$. For each block $b$, we select a (deterministic-then-random) sequence of offsets $\rho_{b,0}, \rho_{b,1}, \cdots \in \{0, \ldots, n-1\}$ with two goals: (i)

ensure systematic coverage of coefficient positions (a deterministic pass with offsets spaced by $m$) and (ii) afterward draw fresh offsets while avoiding repeats until unavoidable, to increase diversity.

For a chosen block $b$ and offset $\rho$, the associated *subsample* $S_{b,\rho}$ is the $n$-row matrix obtained by cyclically rotating the block's coefficient vector by $-\rho$ and applying the sign pattern required by $x^n \equiv -1$ whenever the indices wrap. If $\boldsymbol{v} \in \mathbb{Z}_q^n$ is the coefficient vector of a polynomial $v(x)$ in the block, the rows of $S_{b,\rho}$ are the vectors $\iota(x^j \cdot x^{-\rho} v(x))$ (with wrap-around sign) for $j = 0, \ldots, n-1$. After these $n$ rows, we append the first row again with all coefficients negated, an operation algebraically equivalent to continuing the circulant sequence by one step. This deterministic append increases variability while preserving MLWE relations. Matrices for reduction are built by concatenating subsamples $S_{b,\rho}$ until reaching at least $m$ rows. When assigning blocks to a matrix, we enforce: (a) no block reused within a matrix until all blocks appear, and (b) if $T \geq B$ matrices are built, distinct first blocks are assigned to different matrices. Thus, every row is an image of an original polynomial under a ring automorphism, ensuring algebraic coherence.

**Preparation of the $q$-ary Matrix: Projection Trick and Pruning**. To avoid creating only lattices whose rank is an exact multiple of $n$, a degeneracy that most of the time worsens reduction quality, we construct matrices with $(h+1)n$ rows when the target sample size is $m = hn+g$ with $0 \leq g < n$. After embedding into the standard $q$-ary lattice basis $A_{\text{raw}}$, we introduce a diagonal projector $\Pi$ that zeros the last $n - g$ coordinates of the final circulant block. Algebraically, this projects out coordinates that would otherwise create exact $n$-periodic dependencies. The projected matrix $A_\Pi = \Pi A_{\text{raw}}$ inevitably contains zero rows and zero columns corresponding to the coordinates removed by the projection. Instead of applying an additional reduction step such as LLL to detect dependencies, we prune these trivial rows and columns to obtain a reduced matrix $A_{\text{pruned}}$ of effective size $m$. This guarantees that the active part of the lattice basis has full rank while avoiding unnecessary overhead. After lattice reduction, the pruned zero columns are reinserted, so that the resulting short vectors are embedded back into a space of dimension $(h+1)n$ but remain supported only on the first $m$ positions. This preserves the MLWE structure, improves reduction quality, and ensures that the additional algebraic relations can be exploited in the construction phase.

**Construction of Additional Short Vectors: Negative-Circulant Expansion**. Let $R \in \mathbb{Z}^{t \times (k \cdot n)}$ be the reduced matrix whose rows $r^{(\ell)}$ are short vectors obtained from lattice reduction. Write each row as a concatenation of length-$n$ sub-blocks $r^{(\ell)} = (r_1^{(\ell)} \| r_2^{(\ell)} \| \cdots \| r_L^{(\ell)})$. For each sub-block $r_j^{(\ell)} \in \mathbb{Z}^n$, we form its *negative circulant orbit* $\mathcal{O}(r_j^{(\ell)}) = \{\iota(x^t \cdot r_j^{(\ell)}(x)) : t = 0, \ldots, n-1\}$, where $\iota(x^t \cdot r_j^{(\ell)}(x))$ denotes cyclic rotation and sign flip. Each element of $\mathcal{O}(r_j^{(\ell)})$ has the same Euclidean norm as $r_j^{(\ell)}$, hence preserves shortness. New short vectors are constructed by concatenating rotated sub-blocks in synchrony $\widetilde{r}_t^{(\ell)} = \left(\iota(x^t r_1^{(\ell)}(x)) \| \cdots \| \iota(x^t r_L^{(\ell)}(x))\right)$, giving up to $n$ distinct vectors per $r^{(\ell)}$. Each $\widetilde{r}_t^{(\ell)}$ is a valid relation with respect to the automorphed public matrices $(\sigma_t(A), \sigma_t(b))$, exactly those used in the resampling stage. Thus, the amplification multiplies the number of usable short vectors by roughly a factor $n$ while preserving norm and consistency with the MLWE structure, providing abundant pseudo-samples for the subsequent machine learning step.

## 3.3 Training

After preprocessing, each reduced system of equations can be expressed in the form $(RA, \ Rb = RA \cdot s + Re)$. From the initial $4n$ samples, we generate $l$ different reduced matrices via block-subsampling, and from each matrix we extract up to $t$ short vectors using lattice reduction (the maximum size of the priority queue in our reduction routine determines the bound $t$). The amplification strategy then produces a pool of $l \cdot t \cdot n$ candidate samples after preprocessing. In addition, we attempt to approximate the non-modular samples directly from the distributions of their components (see Appendix E). While this approximation is beneficial in the case of binary secrets, it does not increase the inlier rate for ternary or CBD secrets. Nevertheless, it provides a practical way to estimate the final proportion of inliers, which is otherwise inaccessible since the true non-modular values are unknown. Rather than using all of these samples indiscriminately, we introduce a ranking strategy based on the final estimated standard deviation $\sigma_{Rb}$. Intuitively, a smaller $\sigma_{Rb}$ corresponds to a dataset that is cleaner (fewer modular wrap-arounds and less variance induced by $Re$), even if it is smaller in size. Since in practice we cannot identify outliers directly, we show in Appendix E.1 that minimizing $\sigma_{Rb}$ increases the proportion of inliers and thus raises the probability of recovering the

correct secret. For this reason, we first train the regression model on a suitably chosen subset, large enough to capture the linear pattern $R\mathbf{b} \approx RA \cdot \mathbf{s}$, but restricted enough to avoid the accumulation of outliers.

For the learning phase, we deliberately restrict ourselves to robust linear regression models. Linear models are a natural choice given the underlying algebraic structure of LWE, and their interpretability and efficiency enable us to perform recovery already during preprocessing. We tested multiple robust regressors (see Appendix B) designed to mitigate the effect of corrupted samples by downweighting them during training. Once the regression model converges, the recovery procedure is straightforward: we extract the learned coefficient vector $\hat{\mathbf{s}}$, normalize it with respect to the expected secret distribution, and then round and clip it to enforce the known support of $\mathbf{s}$. We then verify the candidate secret by checking the residual distribution $\mathbf{r} = \mathbf{b} - A\hat{\mathbf{s}}$. If $\mathbf{r}$ is consistent with the known distribution of the original error vector $\mathbf{e}$ (i.e., bounded variance and support), then $\hat{\mathbf{s}}$ is accepted as the correct secret. Otherwise, the candidate is rejected, and training continues on additional subsets of samples until complete recovery.

## 4 EXPERIMENTAL RESULTS

Next, we provide experimental results comparing the *NoMod* approach with the related works.[2] All experiments were executed in parallel on 16 AMD EPYC 7702P CPUs running at 2.00–2.18 GHz. For Kyber parameter sets, we used sparse CBD secrets and CBD errors with $\eta = 2$, while for SALSA, PICANTE, and VERDE comparisons, we considered binary and ternary sparse secrets with Gaussian errors of standard deviation $\sigma = 3$. Our preprocessing pipeline includes a progressive BKZ schedule; however, not all experiments reached block size $40$. In particular, in VERDE settings, we stopped reductions at block size $30$ due to excessive preprocessing time, and for Kyber ($n = 256, k = 3$), we halted at block size $10$. The reported "max samples" entries correspond to the expanded sample pools obtained via our amplification technique; in practice, we used only $75\%$ of those amplified samples for training (and in low-HW instances, $10\%$ of the amplified pool sufficed to recover the secret). We report results in terms of recoverable Hamming weight, estimated rop complexity of a primal uSVP attack, and the reduction factor $\rho_A = \frac{\sigma_{RA}}{\sigma_A}$, which quantifies preprocessing quality.

Importantly, asterisks (*) denote experiments where no target Hamming weight was set, and recovery succeeded against a dense secret; this confirms that *NoMod* is methodologically applicable to the unconstrained CBD distributions used in ML-KEM, whereas the sparse benchmarks are included primarily to ensure fair comparison with baselines that are strictly limited to sparse regimes. We provide detailed ablation studies and additional results in Appendices G and I.

**Attacks on Kyber Parameter Sets**. Table 1 summarizes recovery for Kyber settings with $k = 1, 2, 3$. For RLWE ($k = 1$), we consistently recovered dense secrets up to $n = 120$, and sparse recovery at $n = 128$ with $hw = 56$. Beyond this point, performance drops, with $hw = 9$ at $n = 200$ and $hw = 6$ at $n = 256$, aligning with the exponential growth of reduction cost. For MLWE ($k = 2, 3$), the attack scales less favorably: sparse recovery reaches $hw = 22$ for $n = 64, k = 2$, whereas for dimensions $n \cdot k > 150$, it is limited to $hw \leq 6$. Still, recovery was achieved in cases where the corresponding uSVP hardness estimates exceeded $2^{60}$ rop, highlighting that the attack succeeds well beyond the classical reduction frontier.

**Comparison with SALSA**. Table 2 compares our method against SALSA on binary secrets with Gaussian error. SALSA recovers at most $hw = 4$ for $n \leq 128$, requiring between 1.2 and 68 hours. In contrast, our method scales to much larger weights: e.g., $hw = 25$ at $n = 50$ in 42 seconds, and $hw = 32$ at $n = 64$ in under 4 minutes. Even at $n = 128$, we recover $hw = 8$ in 24 hours, compared to SALSA's $hw = 3$ in 46 hours.

**Comparison with SALSA PICANTE.** Against SALSA PICANTE (Table 3), our attack exhibits similar gains. PICANTE recovers up to $hw = 60$ at $n = 350$ in $\sim$307 hours, while our method recovers the full dense secret ($hw = 175$) in only 17.5 hours. At intermediate dimensions, we consistently outperform: e.g., at $n = 200$, PICANTE achieves $hw = 22$ in 87 hours, while we

---

[2]For details about specific attacks, see Appendix F.

[1]Real operations (rop) needed while performing a primal uSVP attack. Estimated using lattice-estimator https://github.com/malb/lattice-estimator

| $n$ | max $hw$ | $\log_2$ rop[1] | $\log_2$ samples | time |
|---|---|---|---|---|
| | | NoMod | | |
| 32 | 21* | 33.1 | 11.26 | 3 s |
| 40 | 25* | 35.8 | 10.32 | 5 s |
| 50 | 32* | 39.2 | 10.97 | 12 s |
| 64 | 44* | 44.0 | 11.58 | 30 s |
| 70 | 47* | 38.9 | 11.94 | 34 s |
| 80 | 52* | 49.2 | 12.32 | 85 s |
| 90 | 58* | 52.8 | 12.66 | 2 m |
| 100 | 75* | 56.1 | 14.64 | 8 m |
| 110 | 72* | 40.2 | 14.78 | 30 m |
| 120 | 77* | 40.6 | 15.81 | 7 h |
| 128 | 56 | 40.9 | 16.00 | 8.5 h |
| 200 | 9 | 52.0 | 16.29 | 11.5 h |
| 256 | 6 | 60.7 | 16.64 | 10.6 h |

| $n$ | $k$ | max $hw$ | $\log_2$ rop[1] | $\log_2$ samples | time |
|---|---|---|---|---|---|
| | | | NoMod | | |
| 16 | 2 | 21* | 33.1 | 11.26 | 7 s |
| 32 | 2 | 42* | 44.0 | 11.30 | 23 s |
| 40 | 2 | 53* | 49.2 | 11.17 | 2 m |
| 50 | 2 | 60* | 56.1 | 12.46 | 2.3 h |
| 64 | 2 | 22 | 61.9 | 13.20 | 5 h |
| 128 | 2 | 4 | 59.1 | 13.78 | 5 h |
| 256 | 2 | 6 | 105.7 | 14.94 | 40 h |
| 16 | 3 | 32* | 38.7 | 11.31 | 15 s |
| 32 | 3 | 61* | 54.7 | 12.30 | 18 m |
| 40 | 3 | 33 | 60.8 | 13.20 | 5 h |
| 50 | 3 | 9 | 41.2 | 13.20 | 5 h |
| 64 | 3 | 6 | 49.2 | 13.20 | 5 h |
| 128 | 3 | 3 | 78.9 | 14.20 | 5 h |

Table 1: Attack results on CRYSTALS-Kyber settings for RLWE ($k = 1$) and MLWE ($k > 1$).

| $n$ | SALSA | | | | NoMod | | | |
|---|---|---|---|---|---|---|---|---|
| | max $hw$ | $\log_2$ rop[1] | $\log_2$ samples | time | max $hw$ | $\log_2$ rop[1] | $\log_2$ samples | time |
| 30 | 4 | 33.1 | 23.84 | 12.9 h | 15* | 33.1 | 12.06 | 11 s |
| 32 | 3 | 33.1 | 20.93 | 1.2 h | 17* | 33.1 | 12.11 | 15 s |
| 50 | 4 | 36.7 | 25.67 | 49.9 h | 25* | 39.5 | 11.92 | 42 s |
| 64 | 3 | 38.0 | 22.39 | 8 h | 32* | 44.5 | 13.21 | 227 s |
| 70 | 3 | 38.4 | 22.74 | 11.9 h | 35* | 46.5 | 13.21 | 51.1 m |
| 90 | 3 | 39.5 | 23.93 | 43.4 h | 19 | 47.2 | 13.62 | 24.1 h |
| 110 | 3 | 44.1 | 24.07 | 68.8 h | 10 | 50.3 | 13.76 | 24.1 h |
| 128 | 3 | 48.0 | 22.25 | 46.0 h | 8 | 53.6 | 13.94 | 24.1 h |

Table 2: Recoverable success between SALSA and NoMod ML-attack on dataset with $q = 251$ and variable dimension $n$ with binary secret and gaussian error with $\sigma = 3$.

recover $hw = 61$ in 40 hours. In every tested case, our method required significantly fewer samples ($\log_2$ samples $\approx 15$ vs. $\approx 22$ for PICANTE), confirming the effectiveness of our preprocessing and amplification strategy.

| $n$ | $q$ | PICANTE | | | | NoMod | | | |
|---|---|---|---|---|---|---|---|---|---|
| | | max $hw$ | $\log_2$ rop[1] | $\log_2$ samples | time (hours) | max $hw$ | $\log_2$ rop[1] | $\log_2$ samples | time (hours) |
| 80 | 113 | 9 | 46.8 | 22.41 | 42 | 13 | 48.9 | 13.62 | 12.5 |
| 150 | 6421 | 13 | 43.0 | 22.06 | 57 | 31 | 45.3 | 14.21 | 40 |
| 200 | 130769 | 22 | 41.7 | 22.04 | 87 | 61 | 43.3 | 14.94 | 40.1 |
| 256 | 6139999 | 31 | 41.6 | 22.02 | 139 | 104 | 41.8 | 15.21 | 40.3 |
| 300 | 94056013 | 33 | 41.8 | 22.02 | 205 | 87 | 41.9 | 15.43 | 40.8 |
| 350 | 3831165139 | 60 | 42.0 | 22.00 | 307 | 175* | 42.1 | 15.53 | 17.5 |

Table 3: Recoverable success between SALSA PICANTE and NoMod ML-attack on dataset with variable dimensions $n$ and $q$, on binary secrets and gaussian error with $\sigma = 3$.

**Comparison with SALSA VERDE**. Table 4 compares our results with SALSA VERDE across binary and ternary secrets. At $n = 256, q = 3329$, our method achieves comparable recovery ($hw = 7$ vs. 8 binary, 7 vs. 9 ternary), but using only 16 CPUs versus VERDE's thousands of cores. At $n = 256, q = 842779$, we even surpass VERDE in the ternary setting, recovering $hw = 29$ versus 24, while matching binary recovery (31 vs. 33). At $n = 350, q = 1489513$, both methods succeed at similar levels ($hw = 12$ binary and $hw = 11$ ternary for us vs. 12 and 13 for VERDE), again at a fraction of the computational cost. Finally, in the most challenging case ($n = 350, q = 94{,}056{,}013$),

our attack recovers up to $hw = 28$ (binary) and $25$ (ternary), which lags behind VERDE's $hw = 36$ in both cases. However, this gap is explained by VERDE's massive preprocessing effort ($216 \cdot 5000$ CPU hours) compared to our fixed budget of $40 \cdot 16$ CPU hours.

| Attack | s | n | 256 | 256 | 350 | 350 |
|---|---|---|---|---|---|---|
| | | q | 3329 | 842779 | 1489513 | 94056013 |
| VERDE | Binary | Best $h$ | 8 | 33 | 12 | 36 |
| | | $\log_2$ rop[1] | 65.5 | 45.7 | 55.5 | 45.3 |
| | | Recover hrs (1 GPU) | 1.5 | 3 | 1.6 | 1.6 |
| | | Total hrs | 3 | 10.5 | 17.6 | 218 |
| | Ternary | Best $h$ | 9 | 24 | 13 | 36 |
| | | $\log_2$ rop[1] | 66.3 | 45.4 | 55.5 | 45.3 |
| | | Recover hrs (1 GPU) | 3 | 7.5 | 25.6 | 17.6 |
| | | Total hrs | 4.5 | 15 | 41.6 | 234 |
| | | Samples | 4M | 4M | 4M | 4M |
| | | $\rho_A$ | 0.77 | 0.43 | 0.61 | 0.38 |
| | | Preproc. hrs · CPUs | 1.5 · 7812 | 7.5 · 7812 | 16 · 5000 | 216 · 5000 |
| NoMod | Bin. | Best $h$ | 7 | 31 | 12 | 28 |
| | | $\log_2$ rop[1] | 64.9 | 45.7 | 55.5 | 44.8 |
| | Ter. | Best $h$ | 7 | 29 | 11 | 25 |
| | | $\log_2$ rop[1] | 65.0 | 45.7 | 55.3 | 44.7 |
| | | Max samples | 409k | 409k | 560k | 560k |
| | | $\rho_A$ | 0.72 | 0.36 | 0.56 | 0.38 |
| | | Total hrs · CPUs | 40 · 16 | 40 · 16 | 40 · 16 | 40 · 16 |

Table 4: Comparison against SALSA VERDE settings with Gaussian errors ($\sigma = 3$).

## 5 RELATED WORK

The Dual-Hybrid MitM attack targets Decision-LWE with sparse secrets by splitting the public matrix $A = [A_1 \mid A_2]$ and correspondingly $\mathbf{s} = (\mathbf{s}_1, \mathbf{s}_2)$ Howgrave-Graham (2007b). SALSA (*Secret-recovery Attacks on LWE via Seq2Seq with Attention*) is the first end-to-end machine learning attack that targets LWE with small, sparse secrets by training a transformer to operate directly over LWE samples and then converting the trained model into a secret-recovery procedure Wenger et al. (2022). SALSA PICANTE improves on the original SALSA by using lattice-based preprocessing for large-scale data amplification, enabling transformer-based attacks on higher-dimensional LWE instances Li et al. (2023b). SALSA VERDE refines the PICANTE attack by arranging the lattice embedding, optimizing the BKZ preprocessing, and adapting the machine learning pipeline for broader secret distributions Li et al. (2023a). SALSA FRESCA refines the preprocessing pipeline of VERDE by combining the recent FLATTER lattice reduction algorithm with BKZ 2.0 in an interleaved approach, inserting a polishing Charton et al. (2024) step after each iteration to improve basis quality at minimal additional cost Stevens et al. (2024). Cool and the Cruel is a statistical attack where the authors observed that after applying lattice reduction to subsampled LWE matrices, the columns of the reduced matrix $RA$ exhibit sharply varying standard deviations: the first $n_u$ columns (after called the *cruel* bits), retain near-uniform variance $\sigma_u \approx q/\sqrt{12}$. In contrast, the remaining *cool* columns have much smaller variance $\sigma_r \ll \sigma_u$ Nolte et al. (2024). More recently, Wenger et al. provided the first benchmarks for LWE secret recovery on standardized parameters, for small and low-weight (sparse) secrets Wenger et al. (2025).

## 6 SIMULATED ATTACKS ON REAL-WORLD PARAMETERS

To assess the practical implications of our findings, we extend our analysis from sparse secrets and small-scale simulations to cryptographically secure parameters. To this end, we introduce the *NoMod* estimator, a specialized cost model derived from the geometric insights of our attack (see Appendix H for the full derivation and algorithmic details).

We integrated the *NoMod* logic into the standard `lattice-estimator` framework Albrecht et al. (2015) to ensure a fair comparison against established baselines. Table 5 reports the estimated number of operations (in rop) of our approach compared to standard cryptanalytic avenues against the NIST-standardized ML-KEM parameter sets.

| Attack | ML-KEM-512 | ML-KEM-768 | ML-KEM-1024 |
|---|---|---|---|
| Arora-GB | $\approx 2^{\infty}$ | $\approx 2^{\infty}$ | $\approx 2^{\infty}$ |
| BKW | $\approx 2^{178.8}$ | $\approx 2^{238.3}$ | $\approx 2^{315.0}$ |
| uSVP | $\approx 2^{143.8}$ | $\approx 2^{204.9}$ | $\approx 2^{275.1}$ |
| BDD | $\approx 2^{140.2}$ | $\approx 2^{201.0}$ | $\approx 2^{270.7}$ |
| Dual | $\approx 2^{149.9}$ | $\approx 2^{214.3}$ | $\approx 2^{288.5}$ |
| Dual Hybrid | $\approx 2^{139.7}$ | $\approx 2^{196.4}$ | $\approx 2^{262.3}$ |
| BDD MITM Hybrid | $\approx 2^{265.2}$ | $\approx 2^{363.2}$ | $\approx 2^{509.8}$ |
| NoMod (This Work) | $\approx 2^{164.3}$ | $\approx 2^{251.2}$ | $\approx 2^{358.0}$ |

Table 5: Estimated complexity (rop) for various attacks on ML-KEM parameter sets.

The results indicate that for standard, dense parameter sets, the complexity of *NoMod* ($\approx 164$ bits for ML-KEM-512) is strictly higher than classical lattice attacks ($\approx 140 - 144$ bits). This behavior is expected and stems from the geometric trade-offs of our embedding. While standard attacks (like Primal uSVP) target the secret or error vectors directly, our method can theoretically succeed by finding vectors of higher norm. However, the different embedding constructs a lattice with a significantly higher volume than the embeddings used in standard attacks. This volume inflation imposes a severe penalty: to compensate for the massive determinant and still find a vector within the useful norm range, the reduction algorithm must achieve a much lower root Hermite factor ($\delta_0$). Achieving this necessitates a disproportionately high block size ($\beta > 500$), thereby driving the computational complexity above that of standard methods for these dense regimes. Consequently, while *NoMod* is highly effective against non-standard or sparse variants, these estimates confirm that current NIST parameters remain secure against this class of machine learning attacks.

## 7 CONCLUSIONS AND FUTURE WORK

This work introduces a novel cryptanalytic pipeline that integrates a progressive BKZ block size scheduling technique, a strategy for recycling short vectors across reduction tours, and an algebraic amplification method exploiting the structure of Ring-LWE. By analytically optimizing sample usage and lowering the required matrix count, our empirical findings illustrate that these strategies yield tangible advancements over prior machine learning-based attacks. Furthermore, to bridge the gap between experimental observations and cryptographically relevant scales, we implemented a theoretical complexity estimator based on Z-shape simulation and integrated it into the standard lattice-estimator framework, enabling accurate cost predictions for real-world parameters.

A core contribution of this work is the reframing of the cryptographic challenge from a black-box learning task to a white-box statistical inference problem. Prior approaches rely on complex deep learning models that necessitate statistical post-processing to extract secrets from learned representations. In contrast, our approach establishes a direct mapping between learned weights and the coefficients of the secret. This demonstrates that lightweight, interpretable statistical models can perform as effectively as computationally expensive architectures, challenging the assumption that complex self-attention mechanisms are required to capture modular arithmetic structures in LWE.

Furthermore, we emphasize that the preprocessing innovations introduced are model-agnostic. While these optimized datasets could serve as inputs to enhance the convergence of transformer-based architectures, their coupling with robust linear models offers distinct advantages in efficiency. We specifically utilized robust estimators to prioritize sample and computational efficiency, enabling successful recovery with thousands of samples rather than millions typically required. Additionally, the negligible training cost of linear regression facilitates "interleaved recovery," allowing secret extraction attempts to be made directly during the preprocessing phase. This feedback loop is computationally impractical with heavy neural architectures.

In future work, investigating the incorporation of modulo-switching methods, utilized in traditional lattice attacks, might achieve even better efficiency in our pipeline. Another promising direction is to explore the substitution of the classical lattice preprocessing step with machine learning or deep learning-based reduction algorithms. This approach might be viable for machine learning-based attacks, as they do not require strictly exact lattice reductions; rather, approximate short vectors are enough to expose the statistical outliers required for successful secret recovery.

## 8 Data Availability and Ethical Considerations

Assessing the security of post-quantum cryptography represents an important challenge due to the recent standardization of PQC algorithms. Machine learning attacks represent a novel threat that is not well explored. We provide novel mechanisms that help improve the attack performance, which can ultimately allow better assessment of the security of PQC algorithms. We do not do any experiments with human users, so there is no risk of deception. We do not use live systems or violate terms of service, and to the best of our knowledge, we follow all laws. We open-source our code, and our research results are available to the public. Moreover, our research does not contain elements that could potentially negatively impact team members.

## 9 Reproducibility Statement

We provide the source code that includes all algorithms as well as the code to produce the datasets. Appendices provide additional material to understand the attacks and the non-modular approximation.

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

## A  LATTICE REDUCTION TECHNIQUES

**Lenstra–Lenstra–Lovász (LLL) Reduction**  The LLL algorithm, introduced by Lenstra, Lenstra, and Lovász Lenstra et al. (1982), is a polynomial-time lattice reduction algorithm that iteratively applies a *size reduction* step and the *Lovász condition*. The size reduction ensures that each vector $b_i$ is small in the direction of previous vectors:

$$b_i \leftarrow b_i - \sum_{j<i} \mu_{i,j} b_j, \quad \mu_{i,j} = \frac{\langle b_i, b_j^* \rangle}{\|b_j^*\|^2}.$$

Where $b_j^*$ is the Gram-Schmidt orthogonalization of the basis. Instead, the Lovász condition

$$\delta \|b_{i-1}^*\|^2 \leq \|b_i^*\|^2 + \mu_{i,i-1} \|b_{i-1}^*\|^2, \quad 0.25 < \delta \leq 1,$$

controls the success of the LLL-reduction. The output is a reduced basis with the guarantee that the first vector cannot be much larger than the shortest nonzero vector:

$$\|b_1'\| \leq (2/(\sqrt{4\delta - 1}))^{n-1} \cdot \lambda_1(\mathcal{L}).$$

While LLL does not solve SVP exactly, it provides an efficient approximation and forms the foundation for more advanced reduction algorithms, such as BKZ Schnorr & Euchner (1994).

**Block Korkine–Zolotarev (BKZ)**   The BKZ algorithm extends LLL by applying stronger reduction on *blocks* of consecutive basis vectors Schnorr & Euchner (1994). Given a block size parameter $\beta$, BKZ repeatedly selects a $\beta$-dimensional sublattice, projects it onto the orthogonal complement of the preceding vectors, and applies a near-exact SVP solver to this block. The reduced block is then reinserted into the global basis, and LLL is used as a preprocessing and postprocessing step to maintain global size reduction. Increasing $\beta$ improves the quality of the reduced basis, but incurs an exponential increase in runtime.

Heuristically, the quality of a BKZ-reduced basis is described by the *Root Hermite factor* $\delta_0$, which for practical block sizes in the range $50 \leq \beta \leq 1000$ satisfies Chen (2013):

$$\delta_0 \;\approx\; \left( \tfrac{\beta}{2\pi e} \left( \pi\beta \right)^{\frac{1}{\beta}} \right)^{\frac{1}{2(\beta-1)}}. \tag{1}$$

This formula captures the trade-off between increased reduction quality (smaller $\delta_0$) and exponential growth in running time. Moreover, given a lattice of dimension $d$ and volume $\mathrm{Vol}(\mathcal{L})$, the shortest vector length is expected to follow the Gaussian heuristic:

$$\|v_{\min}\| \approx \delta_0^d \cdot \mathrm{Vol}(\mathcal{L})^{1/d}. \tag{2}$$

**BKZ 2.0**   BKZ 2.0 improves the original BKZ with various improvements Chen & Nguyen (2011a):

- Early-abort: limit the number of tours to control runtime.
- Pruned enumeration: skip unpromising branches to speed up SVP searches.
- Improved local preprocessing: better reduction of local blocks before enumeration.
- Optimized initial radius: choose a smaller starting search radius to reduce enumeration effort.

These optimizations significantly improve the efficiency of high-quality lattice reduction without degrading the achieved root-Hermite factor. State-of-the-art implementations follow a running-time model of the form

$$T_{\mathrm{BKZ}}(\beta) \;\sim\; c\, d \cdot t_\beta, \tag{3}$$

with $c = 16$ as empirically calibrated in Albrecht (2017). For the SVP oracle cost $t_\beta$, the best known estimates are

$$t_\beta^{\mathrm{classical}} \approx 2^{0.292\,\beta+16.4}, \qquad t_\beta^{\mathrm{quantum}} \approx 2^{0.265\,\beta+16.4}.$$

This exponential scaling places a premium on carefully tuning $\beta$: too small a block size leads to insufficient reduction, while too large a $\beta$ results in impractical runtimes. Therefore, it is essential in any practical attack to select a block size schedule that balances this trade-off. In progressive BKZ strategies, initially proposed in Chen (2013) and further studied in later works Gama & Nguyen (2008); Schnorr & Shevchenko (2012); Haque et al. (2013); Aono et al. (2016), the block size is increased gradually during the reduction process. This allows for early partial reductions using small $\beta$ values, which in turn accelerate and stabilize subsequent higher-$\beta$ phases.

**Fast Lattice Reduction**   The FLATTER algorithm Ryan & Heninger (2023) offers a high-performance alternative to traditional lattice reduction methods, like LLL and BKZ2.0. It achieves this by using an iterative compression technique that reduces the precision of the lattice basis during each recursive step, thereby accelerating the reduction process without compromising the quality of the reduced basis. This approach allows FLATTER to handle lattices of significantly higher dimensions and bit-lengths than previous algorithms, making it particularly effective for cryptanalytic applications involving large-scale lattices. The algorithm maintains approximation guarantees analogous to LLL, ensuring that the reduced basis remains within a constant factor of the shortest vector. Empirical evaluations demonstrate that FLATTER outperforms existing implementations in terms of speed, especially for lattices with dimensions exceeding 1000 and entries with millions of bits.

## B  REGRESSORS

### B.1  HUBER REGRESSOR

The Huber Regressor Huber (1992) addresses the sensitivity of standard linear regression by combining quadratic and linear loss functions:

$$L(y, \hat{y}) = \begin{cases} \frac{1}{2}(y - \hat{y})^2 & \text{if } |y - \hat{y}| \leq \epsilon, \\ \epsilon|y - \hat{y}| - \frac{1}{2}\epsilon^2 & \text{otherwise.} \end{cases}$$

Residuals smaller than $\epsilon$ follow the squared loss, while larger deviations are linearly penalized, preventing extreme outliers from disproportionately affecting the model. The transition parameter $\epsilon$ is critical: too small, a value risks discarding valid samples as outliers, while too large, a value reduces robustness.

### B.2  RANSAC REGRESSOR

RANSAC (RANdom SAmple Consensus) Fischler & Bolles (1981) is an iterative method that estimates a model from random subsets of data, seeking the one that best fits the most extensive set of inliers. At each iteration, a small random subset is used to fit a provisional model, which is then evaluated against the entire dataset to identify samples whose residuals lie within a fixed tolerance. The model producing the largest consensus set is retained, and its parameters are optionally refined using all inliers. This strategy makes RANSAC exceptionally robust even when outliers corrupt a significant fraction of the training data. The trade-off lies in its higher computational cost compared to direct fitting, due to repeated random sampling and model refitting, particularly in high dimensions.

### B.3  TUKEY'S BIWEIGHT REGRESSOR

Tukey's Biweight regression Chang et al. (2018) implements a bounded influence loss function:

$$L(y, \hat{y}) = \begin{cases} \frac{c^2}{6}\left[1 - \left(\frac{y-\hat{y}}{c}\right)^2\right]^3 & \text{for } |y - \hat{y}| \leq c, \\ \frac{c^2}{6} & \text{otherwise,} \end{cases}$$

where $c$ is a threshold that controls the transition from quadratic to constant loss. While Huber down-weights large residuals linearly, Tukey's Biweight effectively ignores them entirely once they exceed $c$, granting extreme robustness in the presence of higher-magnitude outliers. This robustness makes it particularly effective for datasets where outliers can severely distort parameter estimation, as is the case in our scenario.

## C  FROM RING-LWE TO LWE

Writing each polynomial in $R_q$ in coefficient form as a vector in $\mathbb{Z}_q^n$, multiplication is *negacyclic*: the reduction relation $x^n \equiv -1 \pmod{x^n + 1}$ causes the coefficients to wrap around with a sign inversion. Let

$$a(x) = a_0 + a_1 x + \cdots + a_{n-1} x^{n-1} \quad \longleftrightarrow \quad a = (a_0, a_1, \ldots, a_{n-1})^\mathsf{T} \in \mathbb{Z}_q^n,$$

then multiplication by $a$ in $R_q$ corresponds to multiplication by the *negacyclic* (anti-circulant) matrix

$$\overline{\mathrm{circ}}(a) = \begin{bmatrix} a_0 & -a_{n-1} & -a_{n-2} & \cdots & -a_1 \\ a_1 & a_0 & -a_{n-1} & \cdots & -a_2 \\ a_2 & a_1 & a_0 & \cdots & -a_3 \\ \vdots & \vdots & \vdots & \ddots & \vdots \\ a_{n-1} & a_{n-2} & a_{n-3} & \cdots & a_0 \end{bmatrix} \in \mathbb{Z}_q^{n \times n},$$

whose rows are cyclic shifts of $a$ with the wrapped entries negated. Under the coefficient embedding, a Ring-LWE sample $b(x) = a(x)s(x) + e(x) \bmod q$ is thus equivalent to

$$b = \overline{\mathrm{circ}}(a) \cdot s + e \pmod{q}, \quad b, s, e \in \mathbb{Z}_q^n,$$

which is an LWE instance in $\mathbb{Z}_q^n$ with a highly structured public matrix.

## D    FROM MODULE-LWE TO LWE

Writing each polynomial $a_{ij}$ in coefficient form as an $n$-vector over $\mathbb{Z}_q$, multiplication by $a_{ij}$ corresponds to multiplication by $\overline{\text{circ}}(a_{ij}) \in \mathbb{Z}_q^{n \times n}$. Stacking these blocks yields a structured block-circulant matrix

$$A = \begin{bmatrix} \overline{\text{circ}}(a_{11}) & \overline{\text{circ}}(a_{12}) & \cdots & \overline{\text{circ}}(a_{1\ell}) \\ \overline{\text{circ}}(a_{21}) & \overline{\text{circ}}(a_{22}) & \cdots & \overline{\text{circ}}(a_{2\ell}) \\ \vdots & \vdots & \ddots & \vdots \\ \overline{\text{circ}}(a_{k1}) & \overline{\text{circ}}(a_{k2}) & \cdots & \overline{\text{circ}}(a_{k\ell}) \end{bmatrix} \in \mathbb{Z}_q^{kn \times \ell n}.$$

Let $s \in \mathbb{Z}_q^{\ell n}$ and $e, b \in \mathbb{Z}_q^{kn}$ be the coefficient representations of the secrets, errors, and outputs. Then the MLWE equations become $As + e \equiv b \pmod{q}$. Thus, MLWE is an LWE problem with a highly structured block-circulant matrix.

## E    NON-MODULAR APPROXIMATION

To characterize the distribution of a pre-modular LWE sample $\tilde{b} = As + e$, it is natural and convenient to split the problem into two independent parts. We first approximate the linear contribution $As$, and then we compute moments for the additive noise $e$. Under the standard assumption that the secret $s$ and the error $e$ are independent, the mean and variance of $\tilde{b}$ are the sum of the corresponding contributions:

$$\mu_{\tilde{b},i} = \mathbb{E}[\tilde{b}_i] = \mathbb{E}[(As)_i] + \mathbb{E}[e_i], \qquad \sigma_{\tilde{b},i}^2 = \text{Var}(\tilde{b}_i) = \text{Var}((As)_i) + \text{Var}(e_i). \quad (4)$$

Thus, the central task is to provide explicit expressions for $\mathbb{E}[(As)_i]$ and $\text{Var}((As)_i)$ based on (i) the known row $a_i$ of $A$ and (ii) the distributional law of the coordinates of $s$. We denote by

$$a_i = (A_{i1}, \ldots, A_{in}), \qquad S_1(i) = \sum_{j=1}^n A_{ij}, \qquad S_2(i) = \sum_{j=1}^n A_{ij}^2$$

the row statistics that will appear repeatedly below.

To approximate the distribution of $As$, we begin with the general identities:

$$\mathbb{E}[\tilde{b}_i] = \sum_{j=1}^n A_{ij} \mathbb{E}[s_j], \tag{5}$$

$$\text{Var}(\tilde{b}_i) = \sum_{j=1}^n A_{ij}^2 \text{Var}(s_j) + 2 \sum_{1 \leq j < \ell \leq n} A_{ij} A_{i\ell} \text{Cov}(s_j, s_\ell). \tag{6}$$

They reduce the problem to two ingredients: the per-coordinate moments $\mathbb{E}[s_j]$ and $\text{Var}(s_j)$, and any nonzero covariances $\text{Cov}(s_j, s_\ell)$ which appear when coordinates are coupled (e.g., by fixing the Hamming weight). Below, we compute these quantities for the secret families of interest: Binary, Ternary, and CBD distributions.

1. BINARY SECRET: $s_j \in \{0, 1\}$    We distinguish two common sampling models:

1. Bernoulli (unconstrained): assume $s_j \overset{\text{i.i.d.}}{\sim} \text{Bernoulli}(p)$. Then

$$\mathbb{E}[s_j] = p, \qquad \text{Var}(s_j) = p \cdot (1-p), \qquad \text{Cov}(s_j, s_\ell) = 0 \; (j \neq \ell).$$

Substituting into equation 5–equation 6 gives the closed form

$$\mathbb{E}[(As)_i] = p \cdot S_1(i), \qquad \text{Var}((As)_i) = p \cdot (1-p) \cdot S_2(i). \tag{7}$$

When no Hamming weight constraint is present, this baseline is used (the usual choice is $p = \frac{1}{2}$).

2. Exact Hamming weight $h$: suppose $s$ is sampled uniformly from the set of binary vectors of length $n$ with exactly $h$ ones. In this model, coordinates are exchangeable but no longer independent. Elementary hypergeometric calculations yield

$$\mathbb{E}[s_j] = \frac{h}{n}, \qquad \text{Var}(s_j) = \frac{h}{n}\left(1 - \frac{h}{n}\right), \tag{8}$$

$$\text{Cov}(s_j, s_\ell) = \Pr[s_j = 1, s_\ell = 1] - \left(\frac{h}{n}\right)^2 = \frac{h(h-1)}{n(n-1)} - \left(\frac{h}{n}\right)^2 = -\frac{h(n-h)}{n^2(n-1)} \qquad (j \neq \ell). \tag{9}$$

Inserting equation 8–equation 9 into equation 6 and using $\sum_{1 \leq j < \ell \leq n} A_{ij} A_{i\ell} = \frac{1}{2}\left(S_1(i)^2 - S_2(i)\right)$ produces the compact, exact variance formula

$$\mathbb{E}[(As)_i] = \frac{h}{n} \cdot S_1(i), \qquad \text{Var}((As)_i) = \frac{h(n-h)}{n(n-1)} \cdot \left(S_2(i) - \frac{S_1(i)^2}{n}\right). \tag{10}$$

equation 10 is exact for uniform sampling at fixed weight and is numerically stable: compute $S_1(i)$ and $S_2(i)$ per row and evaluate the prefactor $h(n-h)/(n(n-1))$. The model introduces negative pairwise covariance between coordinates, which reduces the variance of $\langle a_i, s \rangle$. The amount of reduction depends on the concentration of $A$ (through $S_1(i)$ and $S_2(i)$).

2. TERNARY SECRET: $s_j \in \{-1, 0, 1\}$.   As before, we define two common models:

1. Balanced ternary (symmetric): if $P(s_j = -1) = P(s_j = 0) = P(s_j = 1) = 1/3$ then

$$\mathbb{E}[s_j] = 0, \qquad \text{Var}(s_j) = \mathbb{E}[s_j^2] = \frac{2}{3}, \qquad \text{Cov}(s_j, s_\ell) = 0.$$

Hence,

$$\mathbb{E}[(As)_i] = 0, \qquad \text{Var}((As)_i) = \frac{2}{3} S_2(i). \tag{11}$$

2. Exact Hamming weight $h$: when exactly $h$ coordinates are active and each active coordinate is assigned sign $\pm 1$ independently and symmetrically, the per-coordinate mean remains zero and the per-coordinate variance equals the activity probability $p = h/n$. Cross-terms have zero expectation because the sign choices are independent and mean zero, so the covariance contribution vanishes. Thus:

$$\mathbb{E}[(As)_i] = 0, \qquad \text{Var}((As)_i) = \frac{h}{n} S_2(i). \tag{12}$$

3. CENTERED BINOMIAL SECRET ($\text{CBD}_\eta$).   A centered binomial with parameter $\eta$ is generated by summing $\eta$ independent $(+1, 0, -1)$ contributions:

$$s_j = \sum_{t=1}^{\eta} (u_t - v_t), \qquad u_t, v_t \overset{\text{i.i.d.}}{\sim} \text{Bernoulli}(1/2).$$

We again define the two common models:

1. CBD (unconstrained): this yields a symmetric law with

$$\mathbb{E}[s_j] = 0, \qquad \text{Var}(s_j) = \frac{\eta}{2},$$

so that for i.i.d. CBD coordinates

$$\mathbb{E}[(As)_i] = 0, \qquad \text{Var}((As)_i) = \frac{\eta}{2} S_2(i). \tag{13}$$

2. Exact Hamming weight $h$: if one enforces that the final secret has exactly $h$ nonzero coordinates, the per-coordinate variance is reduced by the expected retention probability $\alpha$ of a coordinate. The derivation of $\alpha$ is straightforward:
    (a) Let $P_0 = \Pr[s_j = 0]$ for the raw $\text{CBD}_\eta$. Then the raw nonzero probability is $q = 1 - P_0$.

(b) Let $M$ denote the number of non-zero coordinates among the other $n-1$ positions. Then $M \sim \text{Binomial}(n-1, q)$ and

$$\Pr[M = m] = \binom{n-1}{m} q^m (1-q)^{n-1-m}.$$

(c) Conditioned on $M = m$, when truncating to exactly $h$ non-zeros the current coordinate remains non-zero with probability

$$\begin{cases} 1, & m < h, \\ \dfrac{h}{m+1}, & m \ge h, \end{cases}$$

because ties are resolved uniformly among the currently nonzero positions.

(d) Averaging over $M$ gives the retention probability

$$\alpha = \sum_{m=0}^{h-1} \binom{n-1}{m} q^m (1-q)^{n-1-m} + \sum_{m=h}^{n-1} \binom{n-1}{m} q^m (1-q)^{n-1-m} \cdot \frac{h}{m+1}. \tag{14}$$

Modeling the surviving coordinates as independent with retention probability $\alpha$, the effective per-coordinate variance becomes $\alpha \cdot (\eta/2)$. Hence

$$\mathbb{E}[(As)_i] = 0, \qquad \text{Var}((As)_i) = \left(\frac{\eta}{2} \alpha\right) S_2(i). \tag{15}$$

We now summarize the standard models for the additive error $e$, both of which are used in real-world scenarios and in our experiments. We treat only the moment calculations because the same combination rule equation 4 applies.

1. DISCRETE GAUSSIAN NOISE. If $e_i$ is drawn i.i.d. from a (discrete) Gaussian with mean zero and standard deviation $\sigma_e$, then

$$\mathbb{E}[e_i] = 0, \qquad \text{Var}(e_i) = \sigma_e^2.$$

Thus, the contribution of the error to the total moments is simply additive: for row $i$,

$$\mu_{\tilde{b},i} \leftarrow \mu_{(As),i} + 0, \qquad \sigma_{\tilde{b},i}^2 \leftarrow \sigma_{(As),i}^2 + \sigma_e^2.$$

2. CBD NOISE. If each $e_i$ is drawn i.i.d. from $\text{CBD}_\eta$, then (as for the secret CBD)

$$\mathbb{E}[e_i] = 0, \qquad \text{Var}(e_i) = \frac{\eta}{2}.$$

Again, the error contribution is additive and homogeneous across rows:

$$\sigma_{\tilde{b},i}^2 = \sigma_{(As),i}^2 + \frac{\eta}{2}.$$

The expressions above provide closed formulas for the first two moments of $\tilde{b}_i$ for the secret and error laws used in LWE, including exact treatment of binary fixed-weight covariance and the actual retention factor $\alpha$ for CBD truncation. These moments are the only quantities required to implement the likelihood-ranking used in the selection of candidate unwrapped values.

### E.1 CANDIDATE GENERATION AND EXPECTED INLIER RATE

Given a public observation $b_i \in \mathbb{Z}_q$, candidate integer pre-images are enumerated as $\mathcal{C}(b_i) = \{b_i + kq : k \in \mathbb{Z}\}$ and scored using a Gaussian approximation $\tilde{b}_i \sim \mathcal{N}(\mu_{\tilde{b},i}, \sigma_{\tilde{b},i}^2)$ via the log-likelihood $\ell_i(k) = -(\tilde{b}_k - \mu_{\tilde{b},i})^2/(2\sigma_{\tilde{b},i}^2)$. Restricting candidates to a finite $t$-sigma window yields an upper bound on the set size, $N_{\text{cand},i}(t) \approx \lceil 2t\sigma_{\tilde{b},i}/q \rceil$, and probabilities are normalized across this set as $p_i(k) \propto \exp[-(\tilde{b}_k - \mu_{\tilde{b},i})^2/(2\sigma_{\tilde{b},i}^2)]$. These probabilities either identify the maximum likelihood candidate or propagate a soft-labeled set for downstream refinement. The quality of unreduced LWE samples is quantified by the inlier probability that the correct pre-modular representative corresponds to zero shift, $P_{\text{inlier},i} = \text{erf}(q/2\sqrt{2}\,\sigma_{\tilde{b},i})$, which predicts the fraction of rows immediately recoverable via likelihood maximization. Aggregating over $M$ independent rows, the expected number of inliers is $\mathbb{E}[\#\text{inliers}] = \sum_i P_{\text{inlier},i} \approx M \cdot P_{\text{inlier}}$, providing a practical estimate of recoverable samples prior to any further algorithmic refinement.

# F    ATTACKS ON LWE

## F.1    PRIMAL (USVP) ATTACK

The LWE problem with $(A, b) \in \mathbb{Z}_q^{m \times n} \times \mathbb{Z}_q^m$ samples can be first reduced to a BDD problem, and then Kannan's embedding transforms it into a uSVP instance by solving:

$$B_1 = \begin{pmatrix} B_0 & \mathbf{t} \\ 0 & 1 \end{pmatrix} = \begin{pmatrix} I_n & 0 & 0 \\ A & qI_m & \mathbf{b} \\ 0 & 0 & 1 \end{pmatrix}.$$

The lattice generated by $B_1$ contains the unique shortest vector:

$$\mathbf{v}_{\text{short}} = B_1 \begin{pmatrix} \mathbf{s} \\ \mathbf{c} \\ 1 \end{pmatrix} = \begin{pmatrix} \mathbf{s} \\ \mathbf{e} \\ 1 \end{pmatrix}.$$

If the gap between the shortest vector $\mathbf{v}_{\text{short}}$ and the second shortest vector is sufficiently large, lattice reduction algorithms such as BKZ 2.0 can recover $\mathbf{v}_{\text{short}}$, yielding the secret $\mathbf{s}$. The conditions for recovery are: (i) the secret $\mathbf{s}$ must be small relative to other lattice vectors, (ii) the error $\mathbf{e}$ must satisfy $\|\mathbf{e}\| < \frac{1}{2}\lambda_1(\mathcal{L}(B_0))$ and (iii) the shortest vector in the embedded lattice $B_1$ must be unique. Under these conditions, the uSVP instance derived from LWE via Kannan's embedding guarantees recovery of $\mathbf{s}$.

## F.2    DUAL-HYBRID MEET-IN-THE-MIDDLE (MITM) ATTACK

The Dual-Hybrid MitM attack Howgrave-Graham (2007b) targets Decision-LWE with sparse secrets by splitting the public matrix $A = [A_1 \mid A_2]$ and correspondingly $\mathbf{s} = (\mathbf{s}_1, \mathbf{s}_2)$. A scaled dual lattice on $A_1$ is constructed, and lattice reduction yields short vectors that essentially eliminate the contribution of $\mathbf{s}_1$ to the LWE samples, producing reduced instances depending only on $\mathbf{s}_2$. Repeating this process $\tau$ times generates reduced samples used to build a locality-sensitive hash table of candidate $\mathbf{s}_2$ values. The MitM step finds collisions between guessed and stored candidates, identifying possible partial secrets, which are then verified. The error bound $B$ controls hash sensitivity, and parameters $\tau$, $\zeta$, and $B$ are tuned to balance time, memory, and reduction quality. This attack does not directly recover the whole secret, but efficiently narrows the search space for sparse-secret LWE.

## F.3    SALSA ATTACKS

**SALSA**    SALSA (*Secret-recovery Attacks on LWE via Seq2Seq with Attention*) Wenger et al. (2022) is the first end-to-end machine learning attack that targets LWE with small, sparse secrets by training a transformer to operate directly over LWE samples and then converting the trained model into a secret-recovery procedure. Given many samples $(\mathbf{a}_i, b_i = \langle \mathbf{a}_i, \mathbf{s} \rangle + e_i \bmod q)$ that share the same secret $\mathbf{s}$, SALSA trains a seq2seq transformer to predict $b$ from $\mathbf{a}$, thereby forcing the model to internalize modular linear structure in the presence of noise; the paper first demonstrates that transformers can learn modular arithmetic reliably, then uses this capability for cryptanalysis. After training, SALSA offers two recovery modes: (i) *direct recovery*, which queries the model on carefully chosen inputs so that its outputs reveal coordinates of $\mathbf{s}$; (ii) a *distinguisher-based recovery* mode, in which the trained model serves as an oracle for distinguishing LWE samples from uniformly random ones and this oracle is leveraged to recover the secret via the standard reduction from the search variant of LWE to its decision form; both modes include a verification step that tests candidate secrets by checking that residuals $b_i - \langle \mathbf{a}_i, \hat{\mathbf{s}} \rangle \bmod q$ have slight variance consistent with the noise. Practically, SALSA recovers *sparse binary* secrets in small-to-mid dimensions, but it is sample-hungry (millions of samples in the original experiments) and its effectiveness decreases as dimension or Hamming weight grows.

**SALSA PICANTE**    SALSA PICANTE Li et al. (2023b) improves on the original SALSA by using lattice-based preprocessing for large-scale data amplification, enabling transformer-based attacks on higher-dimensional LWE instances. The core innovation is the preprocessing via an *error-penalized*

*lattice embedding* reminiscent of the dual embedding: for an LWE matrix $A \in \mathbb{Z}_q^{m \times n}$, the algorithm constructs

$$\Lambda = \begin{bmatrix} \omega \cdot I_m & A \\ 0 & q \cdot I_n \end{bmatrix} \tag{16}$$

where $\omega \in \mathbb{Z}$ weights the contribution of error coordinates. Applying BKZ 2.0 to this block matrix yields a unimodular transformation $[R \ C] \in \mathbb{Z}^{(m+n) \times (m+n)}$, producing

$$[R \ C] \cdot \begin{bmatrix} \omega \cdot I_m & A \\ 0 & q \cdot I_n \end{bmatrix} = [\omega \cdot R \quad RA + q \cdot C].$$

This transformation defines a reduced LWE instance $(RA, Rb)$ with transformed errors $Re$. The parameter $\omega$ explicitly mediates the trade-off between minimizing the LWE matrix coordinates and controlling error amplification, enabling a more substantial reduction in $A$ without excessive noise growth.

In practice, PICANTE uses the linearity of LWE to amplify a small number of samples into a synthetic dataset. Starting with only $m = 4n$ real LWE pairs, the algorithm applies a resampling procedure to generate an exponentially large family of $n \times n$ matrices by drawing random subsets of rows. Subsampled matrices initially preserve the original noise distribution and BKZ reduction transforms them via $[R \ C]$, amplifying the error according to $\|R\|$. This process produces millions of reduced LWE samples, which are deduplicated and encoded as token sequences for transformer training.

The secret-recovery phase improves upon SALSA by introducing *cross-attention extraction*, which directly leverages the transformer's attention maps to read secret bits. After training on the multi-million example corpus, the model's attention layers highlight correlations between input tokens and the underlying secret. By systematically interpreting these attention weights, cross-attention extraction can infer individual secret bits with high confidence. This method complements SALSA's direct recovery and distinguisher approaches; combining all three mechanisms yields more accurate secret reconstruction than any single method alone. As a result, PICANTE recovers sparse binary secrets in dimensions up to $n = 350$ with Hamming weights $h \approx n/10$, surpassing SALSA's previous practical limits of $n \leq 128$ and $h \leq 4$.

**SALSA VERDE** SALSA VERDE Li et al. (2023a) refines the PICANTE attack by arranging the lattice embedding, optimizing the BKZ preprocessing, and adapting the machine learning pipeline for broader secret distributions. The first change is the embedding:

$$\Lambda_i' = \begin{bmatrix} 0 & q \cdot I_n \\ \omega \cdot I_m & A \end{bmatrix}. \tag{17}$$

This embedding, while producing a reduced basis of the same form $[\omega \cdot R \ RA + q \cdot C]$, positions $A$ in the lower-right block. By placing $A$ in this block, the rearrangement changes the geometry of the embedding so that the rows containing $A$ are less affected by the error-penalization scaling $\omega$, allowing BKZ to achieve stronger size reduction on them, based on their results. At the same time, the upper block $q \cdot I_n$ is extremely sparse, which reduces the number of nonzero entries processed during size-reduction steps, improving floating-point efficiency. In addition to this structural change, VERDE lowers the penalty parameter from $\omega = 15$ to $\omega = 10$, and incorporates several BKZ engineering optimizations: interleaved reduction, adaptive block size selection, and early stopping. All aimed at cutting down the preprocessing cost. The sample count per embedded matrix is also slightly reduced from $n$ to $0.875n$ without noticeably affecting the attack's success rate.

On the machine learning side, VERDE avoids using a sequence-to-sequence architecture in favor of an encoder-only transformer equipped with rotary position embeddings, trying to capture the cyclic structure of modular arithmetic. In the recovery stage, VERDE drops both the cross-attention and direct recovery modes, relying solely on an improved distinguisher, now extended to a two-bit variant that handles ternary and Gaussian distributions as well. Finally, VERDE attributes many recovery failures for small $q$ to excessive modular wrap-around in the LWE samples. Although the percentage of non-modular samples cannot be measured in practice without knowing $s$, experiments show that successful recovery is strongly correlated with an empirical threshold of about 67%.

**SALSA FRESCA** SALSA FRESCA Stevens et al. (2024) refines the preprocessing pipeline of VERDE by combining the recent FLATTER lattice reduction algorithm with BKZ 2.0 in an interleaved approach, inserting a polishing Charton et al. (2024) step after each iteration to improve basis quality at minimal additional cost. On the machine learning side, FRESCA retains the encoder-only transformer architecture from VERDE, but replaces rotary embeddings with angular embeddings, which represent modular coordinates as points on the unit circle to better capture the inherent periodicity of LWE samples. Furthermore, the attack leverages pre-training on generic LWE-like instances before fine-tuning on the target distribution, substantially reducing the number of task-specific training steps required. These combined optimizations enable efficient recovery of sparse binary secrets in dimensions up to $n = 1024$, extending the reach of the SALSA family to larger parameter regimes.

### F.3.1 THE COOL AND THE CRUEL ATTACK

Cool and the Cruel Nolte et al. (2024) is a statistical attack where the authors observed that after applying lattice reduction to subsampled LWE matrices, the columns of the reduced matrix $RA$ exhibit sharply varying standard deviations: the first $n_u$ columns (after called the *cruel* bits), retain near-uniform variance $\sigma_u \approx q/\sqrt{12}$. In contrast, the remaining *cool* columns have much smaller variance $\sigma_r \ll \sigma_u$. The attack splits recovery into two stages. In the first stage, the small set of cruel bits is brute-forced: for each candidate assignment, one computes residuals $x = \mathbf{a} \cdot s^* - b$ (mod $q$) over the reduced samples. If the cruel assignment is correct, the variance of $x$ remains low; otherwise, it appears nearly uniform, allowing a clear statistical distinction.

Once the cruel bits are fixed, the cool bits are recovered greedily. For each cool coordinate $k$, two hypotheses are tested (bit = 0 or 1) by comparing the variance of residuals under each guess. The correct bit yields a lower variance, enabling linear-time reconstruction of all cool bits. This divide-and-conquer approach dramatically reduces the search complexity and memory requirement compared to full brute force, enabling efficient recovery for dimensions up to $n = 1024$ on moderate hardware Wenger et al. (2025).

## G ABLATION STUDY

To disentangle the contributions of individual pipeline components and quantify their impact on the attack's success, we conducted three targeted ablation studies. These experiments isolate the effects of our vector saving strategy, algebraic sample amplification, and the choice of robust regression algorithms. The results, detailed below, validate our architectural choices and highlight the trade-offs between computational cost and error tolerance.

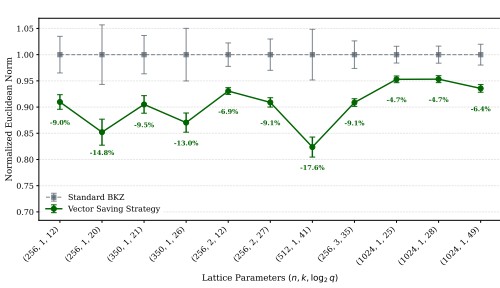
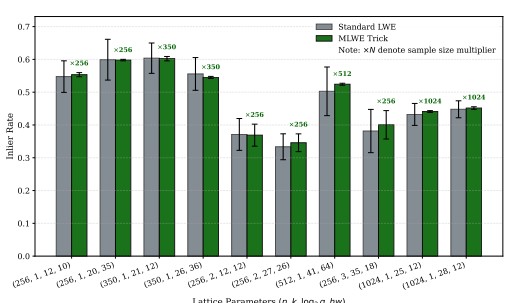

(a) Impact of Vector Saving Strategy

(b) MLWE Optimization Stability

Figure 1: (a) Comparison of the normalized Euclidean norms of the output basis between Standard BKZ and our Vector Saving Strategy. We experiment with matrices $A$ of variable dimensions from 256 to 1024 and various modulo $q$. (b) Comparison of the Inlier Rate between standard LWE samples and those generated via algebraic amplification. The annotation $\times N$ denotes the sample size multiplier. The results show that amplifying the dataset by a factor of $n$ preserves sample quality (inlier rate deviation $< 2\%$).

### G.1 Impact of Vector Saving Strategy

We evaluated the Vector Saving Strategy described in Section 3 across 11 different parameter sets ranging from Ring-LWE low-dimensional instances to Module-LWE high-dimensional samples. By retaining the shortest unique vectors discovered across multiple BKZ tours—rather than discarding them when the basis is updated—we observed a consistent improvement in the quality of the output vectors.

As illustrated in Figure 1, the strategy yields a strictly lower distribution of row norms compared to standard BKZ 2.0 reduction. Specifically, the strategy resulted in an average reduction in the mean row norm of approximately 9.5% across all tested settings, with a maximum observed reduction of 17.6% for the $(512, 1, 41)$ parameter set.

### G.2 MLWE Optimization and Algebraic Amplification

A critical optimization for Ring-LWE and Module-LWE targets is the exploitation of algebraic structure to amplify the training data. We evaluated the trade-off inherent in this technique to verify if the outlier rate remains similar between the original samples and those generated via polynomial rotation.

Our study demonstrates that these algebraic amplification techniques allow us to scale the effective sample pool size by a factor of $n$ (the ring dimension). Figure 1 compares the "Inlier Rate" - the proportion of samples correctly unwrapped to the linear regime - between the standard LWE baseline and our MLWE amplified dataset. Crucially, the massive increase in data quantity does not degrade quality: the inlier rate remains stable, deviating by less than 2% on average compared to the non-amplified baseline. This confirms that rotational expansion provides a valid source of high-quality training data without requiring additional expensive lattice reductions. Instead, the rotation effectively introduces new independent samples into the training set, maximizing the utility of a single lattice reduction.

### G.3 Robustness of Regression Algorithms

Finally, we performed a comparative analysis of the breakdown points for different linear regression algorithms to identify the most resilient estimator for the *NoMod* pipeline. We specifically tested Ordinary Least Squares (OLS), Huber regression, RANSAC, and Tukey's Biweight on subsets of 100,000 samples with varying injected outlier rates.

The results, summarized in Figure 2, reveal distinct performance tiers. Standard RANSAC, often considered the gold standard for robust fitting, performed worse than standalone M-estimators in this high-dimensional setting, likely due to the difficulty of finding a pure inlier set in random sub-sampling. OLS and Huber regression provided moderate robustness but failed at higher noise levels.

In contrast, Tukey's Biweight demonstrated superior resilience, tolerating outlier rates of up to **44%** in some configurations. This robustness incurs a higher computational cost, largely because our implementation relies on `statsmodels` (iteratively reweighted least squares) rather than the highly optimized `scikit-learn` implementations used for OLS and Huber. However, this trade-off is justified: Tukey's Biweight enables our linear model to achieve outlier tolerance comparable to that of state-of-the-art transformer-based attacks like SALSA VERDE and FRESCA, while remaining significantly faster to train.

Furthermore, we investigated advanced interpretability-driven techniques to explicitly filter outliers, including gradient-based detection via cosine similarity of weight updates and unsupervised PCA clustering; however, these heuristics consistently yielded inferior results compared to the implicit outlier rejection provided by Tukey's Biweight loss function.

## H Complexity Estimation

To estimate the attack complexity of our method for any specific parameters, we require a predictive model that accurately estimates the runtime without performing the full reduction. Previous estimators typically rely on the Geometric Series Assumption (GSA) (e.g., Schnorr (2003); Chen &

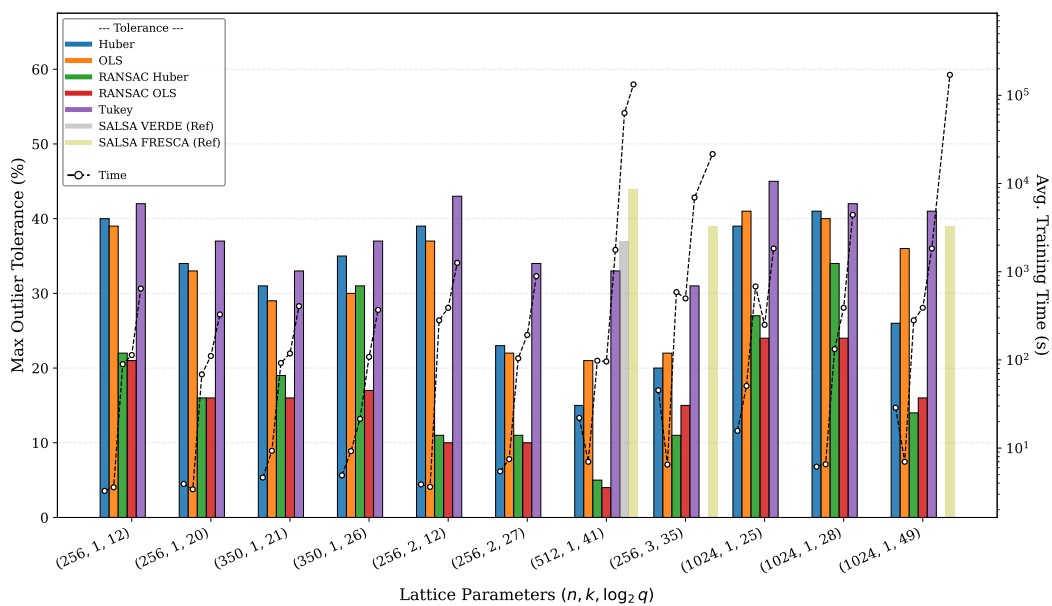

Figure 2: The primary y-axis (bars) displays the maximum outlier tolerance (%) for various regressors across different parameter sets. The secondary y-axis (dashed line) plots the average training time in seconds (log scale). Tukey's Biweight (purple) consistently outperforms OLS, Huber, and RANSAC, matching the robustness of deep learning attacks (grey/yellow bars) at a fraction of the computational cost.

Nguyen (2011b)), which posits a linear decay in the log-norms of the Gram-Schmidt orthogonalized (GSO) basis vectors. However, our attack operates in the pre-asymptotic regime with low block sizes ($\beta$), where the basis profile exhibits a characteristic "Z-shape" rather than a straight line Howgrave-Graham (2007a). Furthermore, the standard GSA fails to capture the nuances of our vector saving strategy, which exploits the variance of reduced vectors rather than just the mean.

Our attack relies on the embedding $\Lambda = \begin{bmatrix} wI_m & A \\ 0 & qI_n \end{bmatrix}$, where the unimodular transformation $[R \quad C]$ yields the reduced basis $[wR \quad RA + qC]$. We propose a refined estimator that separates these contributions to predict the inlier rate $P_{in}$ through $\sigma_{Rb}$.

The following figures all utilize the Kyber parameter set with $q = 3329$, $n = 256$, and $k = 2$.

## H.1 SIMULATING THE GSO PROFILE AND Z-SHAPE

Let $\Lambda$ be the q-ary embedded lattice of dimension $d = m + nk$. Instead of the standard GSA, we utilize the ZGSA (Z-shaped GSA) Ducas & van Woerden (2021) model to simulate the squared Gram-Schmidt norms $\|\mathbf{b}_i^*\|^2$. This model accurately captures the "plateau-slope-plateau" behavior observed in BKZ reductions of q-ary lattices before convergence.

As shown in Figure 3, the simulated GSO profile (dashed line) closely tracks the experimental profile (solid line) of a reduced basis. The profile remains flat for the first indices before entering a decay slope determined by the block size $\beta$.

## H.2 FROM GSO TO EUCLIDEAN ROW NORMS

While the GSO norms $\|\mathbf{b}_i^*\|$ define the orthogonality defect, the attack runtime depends on the actual Euclidean lengths of the row vectors $\|\mathbf{b}_i\|$. We estimate these lengths by modeling the geometry of the basis vectors.

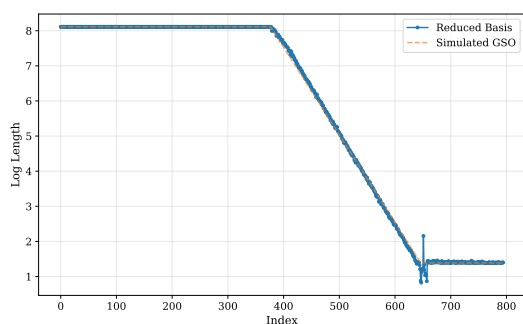

Figure 3: Comparison of the experimental GSO profile (blue) versus the simulated ZGSA profile (orange) for $n = 256, k = 2, q = 3329$. The ZGSA accurately models the pre-asymptotic slope.

For a random reduced basis, the squared length of the $i$-th vector is given by:

$$\|\mathbf{b}_i\|^2 = \|\mathbf{b}_i^*\|^2 + \sum_{j=1}^{i-1} \mu_{i,j}^2 \|\mathbf{b}_j^*\|^2 \tag{18}$$

where $\mu_{i,j}$ are the Gram-Schmidt coefficients.

In the region preceding the slope, we consider the $\|b_i\|$ to be unreduced $q$-vectors that maintain a norm of $q$; conversely, after the start of the slope, we model the coefficients $\mu_{i,j}$ as uniformly distributed random variables in $[-0.5, 0.5]$, with $\mathbb{E}[\mu^2] = 1/12$.

A crucial component of our attack is the vector saving strategy, which recycles short vectors across multiple tours. We are not limited to the average behavior of the basis; we select the best candidates from the distribution of found vectors. To model this, we calculate not only the expected length $\mathbb{E}[\|\mathbf{b}_i\|]$ but also the standard deviation $\sigma(\|\mathbf{b}_i\|)$ induced by the randomness of $\mu_{i,j}$. We derive the variance of the squared lengths using the variance of the squared coefficients, $\mathrm{Var}(\mu^2) = 1/180$:

$$\mathrm{Var}(\|\mathbf{b}_i\|^2) \approx \sum_{j=1}^{i-1} \mathrm{Var}(\mu_{i,j}^2)\|\mathbf{b}_j^*\|^4 = \frac{1}{180} \sum_{j=1}^{i-1} \|\mathbf{b}_j^*\|^4 \tag{19}$$

To obtain the standard deviation of the Euclidean length itself, we apply the Delta method approximation $\sigma(Y) \approx \frac{\sigma(X)}{2\sqrt{\mathbb{E}[X]}}$ for $Y = \sqrt{X}$:

$$\sigma(\|\mathbf{b}_i\|) \approx \frac{\sqrt{\mathrm{Var}(\|\mathbf{b}_i\|^2)}}{2 \cdot \mathbb{E}[\|\mathbf{b}_i\|]} \tag{20}$$

We estimate the minimum accessible norm as the lower bound of this distribution: $\|\mathbf{b}_{min}\| \approx \mathbb{E}[\|\mathbf{b}_i\|] - 2.5 \cdot \sigma(\|\mathbf{b}_i\|)$. This shift accounts for the ability of our strategy to find outliers in the reduction distribution.

Figure 4 shows that the simulated row norms (dashed), adjusted by $\pm 2.5\sigma$ bounds, successfully bracket the experimental minimum row norms found by our pipeline.

### H.3 COMPONENT SEPARATION

To calculate the final error variance $\sigma_{Rb}^2$, we rely on the Euclidean norms of the individual row vectors within the transformation matrix $R$ (the left part of the reduced basis), which amplify the error $e$, and the reduced matrix $RA + qC$ (the right part), which amplifies the secret $s$. Thus, the total variance is modeled as $\sigma_{Rb}^2 = \|R_i\|^2\sigma_e^2 + \|RA_i + qC_i\|^2\sigma_s^2$, where $\|R_i\|$ and $\|RA_i + qC_i\|$ denote the norms of the corresponding rows.

In standard analyses, it is often assumed that the row norm is uniformly distributed across columns. However, consistent with observations in the "Cool and the Cruel" attack Nolte et al. (2024), the variance is not distributed evenly. Indeed, recent analysis demonstrates that this phenomenon is an

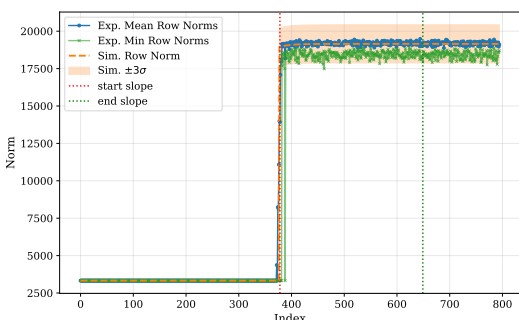

Figure 4: Simulated row norms versus experimental results. The vector saving strategy allows us to access vectors (green crosses) significantly shorter than the mean reduced basis vectors (blue dots), aligning with the lower bound of our simulation.

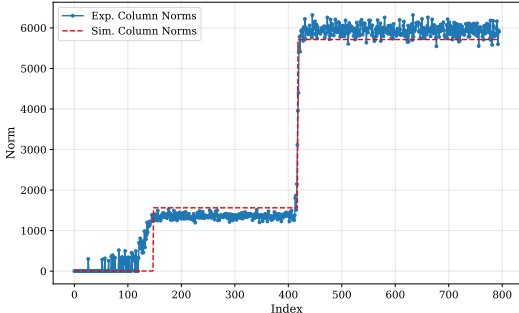

Figure 5: Comparison of experimental and simulated column norms of the reduced basis. The simulated profile (dashed red line) accurately tracks the experimental values (solid blue line), validating the analytical model.

expression of the geometry of the well-known Z-shape basis in $q$-ary lattices Karenin et al. (2025). As shown in Figure 5, the column norms exhibit a distinct stair-step pattern. We estimate this profile analytically by processing the simulated Gram-Schmidt norms $\beta_i = \|\mathbf{b}_i^*\|^2$ from the ZGSA.

We classify the basis indices into three regions based on the embedding parameters $\omega$ (penalty) and $q$ (modulus):

$$N_{\text{id}} = \|i \mid \beta_i \leq \omega^2\| \tag{21}$$

$$N_{\text{cruel}} = \|i \mid \beta_i \geq q^2\| \tag{22}$$

$$N_{\text{slope}} = d - N_{\text{id}} - N_{\text{cruel}} \tag{23}$$

We define a geometry scaling factor $\gamma = \sqrt{d/N_{\text{slope}}}$ to account for the mixing of vectors, which effectively concentrates in the slope region. The predicted column norm profile $\|\mathbf{c}_j\|$ is then constructed as the following system of equations:

$$\|\mathbf{c}_j\| \approx \begin{cases} \omega & 1 \leq j \leq N_{\text{id}} \\ \gamma \cdot \sqrt{\frac{1}{N_{\text{slope}}} \sum_{\omega^2 < \beta_k < q^2} \beta_k} & N_{\text{id}} < j \leq N_{\text{id}} + N_{\text{slope}} \\ \gamma \cdot q & N_{\text{id}} + N_{\text{slope}} < j \leq d \end{cases} \tag{24}$$

This modeled profile allows us to compute the overall weight contribution of the first $m$ columns versus the remaining $n$ columns. By partitioning the simulated row norms based on the weights ratio, we obtain independent estimators for $\|R_i\|$ and $\|RA_i + qC_i\|$. Figure 6 validates this approach, showing that our simulation accurately tracks the separate growth of both components.

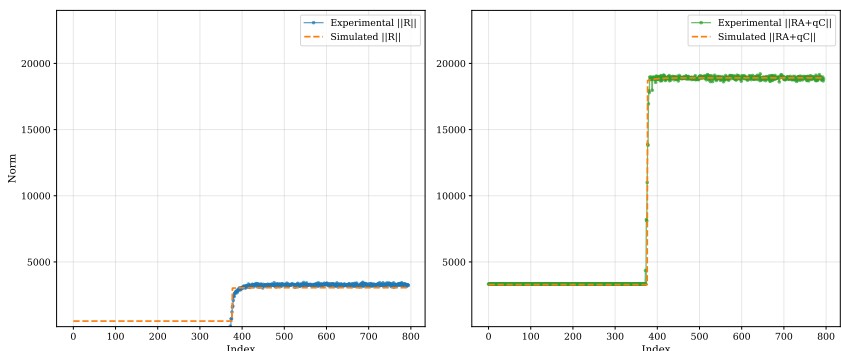

Figure 6: Component-wise norm estimation. Left: Estimation of the row norms of $\|R\|$ (error amplification). Right: Estimation of the row norms of $\|RA + qC\|$ (secret amplification). The simulation (orange) closely predicts the experimental values (blue/green).

## H.4  Final Cost

Finally, to predict the success of the attack, we select the vector index corresponding to the start of the slope in the GSO profile as the optimal extraction point. Using the simulated values for $\|R_i\|$ and $\|RA_i + qC_i\|$ at this index, we compute the standard deviation of the reduced sample $\sigma_{Rb}$. The predicted inlier rate is then derived via the error function $P_{in}(\beta) = \text{erf}\left(\frac{q}{2\sqrt{2}\sigma_{Rb}}\right)$ (Section E.1). We determine the minimum block size $\beta$ required to reach a target $P_{in}$ (e.g., sufficient for the robust estimator to converge) and estimate the runtime cost as $T \approx 2^{0.292\beta}$.

## I  Additional Benchmarks

In addition to the comparisons against SALSA, PICANTE, and VERDE, we also benchmarked our implementation against the recent study by Wenger et al. (2025). We include these experiments separately because, in most cases, our method still underperforms compared to the state-of-the-art attacks. Nevertheless, they provide a useful perspective on the trade-off between recoverable Hamming weight and computational resources.

Table 6 shows that our attack successfully recovered sparse secrets in regimes where classical uSVP-based attacks fail altogether, thereby demonstrating the effectiveness of our preprocessing approach. Notably, for $(n, k, q) = (256, 2, 3329)$, we achieved $hw = 6$, surpassing the MitM baseline ($hw = 4$). This indicates that even with significantly fewer computational resources, our method can outperform certain combinatorial strategies in practical settings. However, for higher moduli and larger $k$, our recovered weights remain below those obtained by SALSA FRESCA and the Cool and the Cruel (e.g., $hw = 8$ vs. 18 at $q = 179067461$, and $hw = 6$ vs. 19 at $q = 34088624597$). We emphasize, however, that these competing attacks rely on massive parallelism or large memory budgets. For instance, FRESCA requires 256 GPUs for secret recovery with up to 3216 CPUs used for preprocessing in the $(256, 2, 3329)$ setting, while The Cool and the Cruel (CC) demands 256 GPUs for recovery with up to 161 CPUs used during preprocessing, making both attacks highly CPU- and GPU-intensive. In contrast, the MitM attack is primarily memory-expensive, requiring over 3.3 TB for $(256, 2, 179067461)$ and $> 42$ TB for $(256, 3, 34088624597)$. On the other hand, our implementation was designed to prioritize efficiency, completing each experiment within a fixed budget of 16 CPU cores and without GPU acceleration.

| Attack | | Kyber MLWE Setting $(n, k, q)$ | | |
|---|---|---|---|---|
| | | (256, 2, 3329) binomial | (256, 2, 179067461) binomial | (256, 3, 34088624597) binomial |
| uSVP | Best $h$ | - | - | - |
| | Recover hrs (1 CPU) | $> 1100$ | $> 1100$ | $> 1100$ |
| FRESCA | Best $h$ | 9 | 18 | 16 |
| | $\log_2$ rop[1] | 108.3 | 58.1 | 67.3 |
| | $\log_2$ samples | 20.93 | 20.93 | 20.93 |
| | $\rho_A$ | 0.88 | 0.67 | 0.69 |
| | Preproc. hrs $\cdot$ CPUs | $28 \cdot 3216$ | $11 \cdot 3010$ | $23 \cdot 1843$ |
| | Recover hrs $\cdot$ GPUs | $8 \cdot 256$ | $16 \cdot 256$ | $6 \cdot 256$ |
| | Total hrs | 36 | 27 | 39 |
| CC | Best $h$ | 11 | 25 | 19 |
| | $\log_2$ rop[1] | 110.0 | 58.7 | 67.5 |
| | $\log_2$ samples | 16.61 | 16.61 | 16.61 |
| | $\rho_A$ | 0.88 | 0.67 | 0.69 |
| | Preproc. hrs $\cdot$ CPUs | $28 \cdot 161$ | $11 \cdot 151$ | $23 \cdot 92$ |
| | Recover hrs $\cdot$ GPUs | $0.1 \cdot 256$ | $42 \cdot 256$ | $0.9 \cdot 256$ |
| | Total hrs | 28.1 | 53 | 34 |
| NoMod | Best $h$ | 6 | 8 | 6 |
| | $\log_2$ rop[1] | 105.2 | 57.0 | 66.1 |
| | $\log_2$ max samples | 17.64 | 17.64 | 18.64 |
| | $\rho_A$ | 0.81 | 0.61 | 0.64 |
| | Prep. + rec. hrs $\cdot$ CPUs | $40 \cdot 16$ | $40 \cdot 16$ | $40 \cdot 16$ |
| MiTM (Decision-LWE) | Best $h$ | 4 | 12 | 14 |
| | $\log_2$ rop[1] | 101.9 | 57.6 | 67.2 |
| | Memory requirements | 10MB | $> 3.3$TB | $> 42$TB |
| | Preproc. hrs $\cdot$ CPUs | $0.5 \cdot 50$ | $1.6 \cdot 50$ | $4.4 \cdot 50$ |
| | Decide hrs (1 CPU) | 0.2 | 0.01 | 25 |
| | Total hrs | 0.7 | 1.61 | 29.4 |

Table 6: Benchmark against existing machine learning-based attacks on real-world settings of CRYSTALS-Kyber. Details about other attacks taken from Wenger et al. (2025).

