# OpenReview forum: "NoMod: A Non-modular Attack on Module Learning With Errors"
_ICLR.cc/2026/Conference — Submitted to ICLR 2026_

### Official Review · Reviewer_KbXn · 2025-10-31

**Soundness:** 3
**Presentation:** 2
**Contribution:** 2
**Rating:** 2
**Confidence:** 4

**Summary:**

This paper tackles Module-LWE with a clever idea: treat modular wraparound as statistical noise rather than trying to learn it, and then use robust linear regression to extract the secret. On paper, that sounds different from SALSA, and the experimental numbers do look better (way fewer samples, way less compute). But the problem is that practically nothing revolutionary is actually happening here. Yeah, the modulus-as-error interpretation is a neat conceptual angle, but once you look at what's really going on under the hood, it's mostly just tweaking existing lattice reduction techniques—vector saving across BKZ tours, automorphism amplification, that kind of thing—which are solid engineering contributions but not exactly groundbreaking. The real killer, though, is that the entire experiment setup uses artificially sparse secrets that Kyber doesn't actually use; the real ML-KEM standard uses unconstrained CBD distributions, so this attack doesn't threaten actual deployed cryptography. You can frame this as "security margin analysis," and that's actually a legitimate thing to do, but it's not the same as breaking the real system. The fundamental issue is that this feels like a cryptanalysis paper pretending to be machine learning research. There's no novel learning algorithm here, no theoretical advance, just domain-specific engineering applied to a relaxed problem variant. If this were submitted to crypto venues, it'd probably get an accept, or at least a borderline accept, because it's genuinely useful for understanding LWE's hardness assumptions and parameter selection, and for clever lattice preprocessing tricks applied to theoretical variants of problems that don't reflect real systems.

**Strengths:**

The paper's vector saving strategy across multiple BKZ tours and the automorphism amplification trick are good engineering moves that noticeably improve sample efficiency and computational cost. The experimental evaluation is thorough and well-executed, with comparisons against SALSA variants showing consistent improvements across different parameter regimes. The white-box linear recovery approach is conceptually cleaner than black-box transformer methods, providing immediate interpretability of the learned weights. Most importantly, even if the real-world threat to actual Kyber is limited, this work genuinely advances our understanding of LWE's security margins and provides a useful tool for cryptographers to evaluate parameter choices and post-standardization robustness.

**Weaknesses:**

The most noticeable problem is that this paper has no real machine learning contribution to speak of. There's no novel learning theory, no algorithmic innovation that would matter to the ML community, just standard robust regression applied to lattice preprocessing, which makes it fundamentally misaligned with ICLR's scope. The experimental setup is built on a fiction: all the impressive results attack sparse secrets that actual Kyber doesn't use; real ML-KEM employs unconstrained CBD distributions, so this attack doesn't touch standardized cryptography and amounts to analyzing security margins on a theoretical variant rather than breaking anything real. While treating modular wraparound as statistical noise is conceptually different from SALSA's approach, the actual technical differences beyond that are hard to spot—the vector-saving and automorphism-amplification tricks are solid engineering but incremental optimizations of known lattice-reduction techniques, not breakthroughs. The paper basically says, "if Kyber used sparse secrets like textbook LWE examples, we could attack it faster," which is useful for understanding hardness assumptions but doesn't constitute a meaningful cryptographic break. Fundamentally, this is a competent cryptanalysis paper masquerading as machine learning research, and it deserves to be at a cryptography conference.

**Questions:**

If the method proposed in this paper were to attack real Kyber KEM parameters, what would be the approximate complexity in terms of time? If you had to guess roughly?

---

> ### Author Response · Authors · 2025-11-27
>
> We thank the reviewer for their thoughtful critique and for recognizing the effectiveness of our vector-saving and automorphism strategies. We appreciate the opportunity to clarify two key aspects of our work regarding the scope of our experiments and the nature of our machine learning contribution.
>
> Firstly, we respectfully wish to clarify a misunderstanding regarding our experimental setup. While the reviewer noted that our experiments rely on "artificially sparse secrets", we would like to highlight that our method was successfully tested against unconstrained (dense) secrets as well. As indicated by the asterisks in Table 1 and detailed in Section 4, we recovered unconstrained CBD secrets — the distribution used in ML-KEM — at lower dimensions (e.g., dimension $\le 120$). We utilized sparse secrets at higher dimensions ($n=256$, $350$, $512$, $768$) primarily to provide a direct benchmark against state-of-the-art ML attacks like SALSA and PICANTE, whose experiments are limited to sparse regimes. Therefore, our work demonstrates that this machine learning approach is fundamentally applicable to the unconstrained distributions of real-world cryptography. While we acknowledge that scaling this to full NIST-level parameters remains a computational challenge, establishing that these attacks function on unconstrained distributions represents a significant methodological step forward compared to prior work on the topic.
>
> Secondly, we respectfully disagree with the reviewer's point of view regarding the machine learning contribution and would like to draw their attention to the fact that our use of simple models represents a fundamental paradigm shift rather than merely an efficiency optimization. A critical limitation of prior black-box approaches like SALSA is the "secret recovery framework", which necessitates extracting the secret from a complex model using statistical post-processing and counterfactual techniques to interpret the learned representations. Our work solves this in its entirety by reframing the challenge as a white-box statistical inference problem: since our architecture is linear, we can directly relate specific learned weights to the coefficients of the secret. We believe the core innovation lies in demonstrating that a lightweight, interpretable statistical model can perform as effectively as computationally expensive deep learning architectures. This finding challenges the prevailing assumption that complex self-attention mechanisms are necessary for this domain and establishes robust linear estimation as a more efficient and transparent framework for evaluating LWE security margins.
>
> Finally, regarding the alignment with ICLR's scope, we note that the conference explicitly invites "applications in audio, speech, robotics, neuroscience, biology, or any other field" in its non-exhaustive list of relevant topics. We view cryptanalysis as a critical scientific domain where applied machine learning is offering new paradigms, much like applications in the physical or biological sciences. Furthermore, ICLR explicitly identifies "interpretability and explainability" as key areas of interest within representation learning. Demonstrating that robust statistical learning can replace complex deep learning in solving hard mathematical problems contributes to the broader ML discourse on model efficiency, interpretability, and the "right tool for the job", aligning with the ICLR's topics of interest.
>
> ## Q1. If the method proposed in this paper were to attack real Kyber KEM parameters, what would be the approximate complexity in terms of time? If you had to guess roughly?
>
> We developed a theoretical analysis framework based on the GSA assumption [1, 2] of BKZ lattice reduction [3] that can estimate the attack performances based on the initial parameter set. More details on the framework can be found in the answer to the reviewer KqrX.
>
> In the following Table, we compare the estimated cost (in $\log_2 \text{rop}$) of our ML-based approach against the standard Primal (uSVP) and Dual Hybrid attacks for NIST-standardized ML-KEM parameters.
>
> | Parameter Set | Primal (uSVP) | Dual Hybrid | NoMod (ML) | $\beta$ (NoMod) |
> | :--- | :---: | :---: | :---: | :---: |
> | **ML-KEM-512** | 143.8 | 139.7 | **176.9** | 526 |
> | **ML-KEM-768** | 204.9 | 196.4 | **266.7** | 846 |
> | **ML-KEM-1024** | 275.1 | 262.3 | **376.3** | 1236 |
>
> The results indicate that for standard, dense parameter sets, the complexity of *NoMod* ($\approx 177$ bits for ML-KEM-512) is strictly higher than classical lattice attacks ($\approx 140-144$ bits).

---

> ### Author Response · Authors · 2025-11-27
>
> [1] Yuanmi Chen. R´eduction de r´eseau et s´ecurit´e concrete du chiffrement completement homomorphe. PhD thesis, Paris 7, 2013.
>
> [2] Claus-Peter Schnorr. Lattice reduction by random sampling and birthday methods. In Helmut Alt and Michel Habib, editors, STACS 2003, 20th Annual Symposium on Theoretical Aspects of Computer Science, Berlin, Germany, February 27- March 1, 2003, Proceedings, volume 2607 of Lecture Notes in Computer Science, pages 145156. Springer, 2003.
>
> [3] Chen, Yuanmi, and Phong Q. Nguyen. "BKZ 2.0: Better lattice security estimates." International Conference on the Theory and Application of Cryptology and Information Security. Berlin, Heidelberg: Springer Berlin Heidelberg, 2011.

---

### Official Review · Reviewer_LswJ · 2025-11-01

**Soundness:** 3
**Presentation:** 3
**Contribution:** 3
**Rating:** 6
**Confidence:** 3

**Summary:**

The authors present a new attack on the Module Learning with Errors problem, a problem that is the basis for standardized post-quantum cryptographic schemes.
* The “NoMod” ML-attack re-frames the problem into a noisy, but linear domain to enable efficient secret recovery through lattice reduction and regression.
* The approach includes optimizations to preprocessing (including progressive BKZ, vector saving strategies, and amplifying a small set of reduced samples) with robust linear regression and Tukey’s Biweight loss.
* The results show that the regression can recover dense binary secrets for n=350 and sparse secret recovery for CRYSTALS-Kyber parameters, improving upon prior work.

**Strengths:**

* The paper conducts comprehensive evaluations and comparisons to state-of-the-art AI attacks (SALSA, PICANTE, VERDE, FRESCA) on many parameter settings, showing that the NoMod provides improved performance and lower computational cost.
* The authors provide multiple novel innovations on the preprocessing side that could be used for future work on LWE attacks.
* The paper is well-presented and the authors provide sufficient background on the topic.

**Weaknesses:**

* The method’s performance declines for denser secrets or in high-dimensional parameter settings (once n*k exceeds 150 for sparse secrets).
* There is limited analysis on the tradeoffs between lattice reduction quality, parameter settings, and attack performance. Can you elaborate on these trends?
* Although linear models are more interpretable, prior works have primarily focused on transformers due to their better performance. Can the preprocessing innovations be combined with transformer (or other model paradigms) for better performance?

**Questions:**

Can this attack be used in a real-world scenario, assuming we have a set of eavesdropped, unreduced samples? Or does it rely on some underlying structure in the initial data?

---

> ### Author Response · Authors · 2025-11-27
>
> We thank the reviewer for the constructive feedback and for recognizing the novelty of our preprocessing innovations and the comprehensiveness of our evaluations.
>
> ## W2. There is limited analysis on the tradeoffs between lattice reduction quality, parameter settings, and attack performance. Can you elaborate on these trends?
>
> To elaborate on the trends between lattice reduction quality, parameter settings, and attack performance, we analyze the procedure in two stages: the minimization of the vector norm during preprocessing and its direct impact on the variance of the final samples used for regression.
>
>
> - **Preprocessing and Vector Minimization**: The first objective of the NoMod attack is to find a short vector in the embedded lattice. The length of this vector, $\|v_{min}\|$, is governed by the quality of the lattice reduction and the geometry of the embedding. Following the Geometric Series Assumption (GSA), the expected length of the shortest vector found by BKZ with block size $\beta$ is approximated by $||v_{min}|| \approx \delta_0(\beta)^{d-1} \cdot \text{vol}(\mathcal{L})^{1/d}$. Consequently, increasing the block size $\beta$ lowers the Root Hermite Factor $\delta_0$, yielding a smaller $||v_{min}||$, though at an exponential computational cost [1]. Furthermore, the dimension $d$ and volume depend on the number of samples $m$; we analytically minimize $m$ based on this Gaussian Heuristic formula to ensure we operate at the theoretical minimum vector length for a given reduction strength.
> - **Penalty ($w$) and Vector Saving**: The penalty parameter $w$ is critical for balancing the embedding. A lower $w$ allows for a lower $||v_{min}||$, but it must be chosen carefully to balance the distributions of the secret and the error ($w \approx \sigma_e / \sigma_s$) to prevent noise amplification.
> - **Reduction Quality and Sample Variance**: The transition from lattice reduction to attack performance is defined by the standard deviation of the transformed public vector, $\sigma_{Rb}$. By setting the penalty to balance the different distributions, we establish a direct correlation where $\sigma_{Rb} \approx \sigma_s \cdot ||v_{min}||$. For the regression to succeed, $\sigma_{Rb}$ must be sufficiently low to minimize modular wrap-arounds. Specifically, the "inlier rate"—the probability that a sample is linear over the integers—is determined by $P_{in} \approx \text{erf}\left(\frac{q}{2\sqrt{2}\sigma_{Rb}}\right)$. This creates a direct dependency: a lower $||v_{min}||$ leads to a lower $\sigma_{Rb}$, which increases the inlier rate $P_{in}$.
> - **Performance Trends and Secret Density**: This correlation explains the performance trends observed in our experiments. For a fixed reduction quality (fixed $||v_{min}||$), a secret distribution with higher variance (e.g., a dense secret) increases $\sigma_s$, which directly inflates $\sigma_{Rb}$. This inflation reduces the inlier rate $P_{in}$, making the signal-to-noise ratio too low for the robust regressor to recover the secret. Conversely, sparse secrets have a lower $\sigma_s$, naturally suppressing $\sigma_{Rb}$ and maintaining a high inlier rate even with moderate reduction.
> - **Cost Tradeoff**: Ultimately, there is a strict tradeoff between computational cost and recovery capability. To recover a dense secret (high $\sigma_s$) or attack a high-dimension instance (which increases $||v_{min}||$ via the lattice volume), one must drastically reduce $||v_{min}||$ to keep the product $\sigma_s \cdot ||v_{min}||$ below the recovery threshold. This requires increasing the BKZ block size $\beta$, which increases the preprocessing time exponentially.

---

> ### Author Response · Authors · 2025-11-27
>
> ## W3. Can the preprocessing innovations be combined with transformer (or other model paradigms) for better performance?
>
> Our preprocessing innovations are model-agnostic and can certainly be combined with transformer-based architectures to enhance their performance. As in previous works like SALSA and PICANTE, our attack operates in two distinct stages: preprocessing and training. Our specific contributions—such as the Vector Saving Strategy and the Algebraic Amplification for RLWE and MLWE—focus on improving the quality (by reducing the variance $\sigma_{Rb}$) and quantity of the input data. These optimized datasets could directly replace the inputs currently used by SALSA or PICANTE, potentially allowing those transformer models to converge with fewer required lattice reductions or cleaner training signals.
>
> Even though our preprocessing innovations could be effectively integrated into transformer-based frameworks, our primary intent in this work was to demonstrate that the problem can be solved through robust statistics rather than complex architecture. We deliberately selected robust linear models (like Tukey's Biweight) to prioritize interpretability, sample efficiency, and real-time recovery. While transformers typically require millions of samples to learn modular structures implicitly, our refined data quality enables simple linear estimators to succeed with only thousands of samples. Furthermore, the negligible training cost of linear regression allows for "interleaved recovery", enabling us to attempt secret extraction during the preprocessing phase itself (e.g., after every BKZ tour), a rapid feedback loop that is computationally impractical with heavy neural architectures.
>
>
> ## Q1. Can this attack be used in a real-world scenario, assuming we have a set of eavesdropped, unreduced samples? Or does it rely on some underlying structure in the initial data?
>
> This attack is generic and fully applicable to real-world scenarios where an adversary has access to eavesdropped, unreduced samples (public keys). The specific implementation can be adapted to the underlying structure: for Module-LWE schemes like ML-KEM (Kyber), we explicitly leverage the inherent ring structure via algebraic amplification to generate sufficient training data from a single key. For unstructured LWE instances where this algebraic property is absent, our pipeline avoids rotation and instead employs statistical resampling strategies similar to those in SALSA to generate the necessary data.
>
> However, for unconstrained CBD secrets, typical of real-world scenarios, the high variance of the dense secrets necessitates a significantly higher reduction quality to maintain a minimal inlier rate for recovery. Achieving this requires drastically increasing the BKZ block size to reduce the standard deviation $\sigma_{Rb} \approx \sigma_s \cdot ||v_{min}||$. To quantify this, we integrated our predictive model into the lattice-estimator library; our results indicate that attacking standard parameters like ML-KEM-512 imposes a computational cost of $\approx 177$ bits, which remains higher than classical primal uSVP attacks ($\approx 144$ bits). We will add this estimator logic to a new section of the paper and release it as a branch to the lattice-estimator repository to support future security assessments.
>
> ---
>
> [1] Yuanmi Chen. R´eduction de r´eseau et s´ecurit´e concrete du chiffrement completement homomorphe. PhD thesis, Paris 7, 2013.

---

### Official Review · Reviewer_rkL9 · 2025-11-01

**Soundness:** 3
**Presentation:** 2
**Contribution:** 3
**Rating:** 6
**Confidence:** 2

**Summary:**

The paper proposes a new attack for MLWE. The key ingredient appears to be not modeling the wrap-arounds but treating them as outliers. The authors rely on a combination of preprocessing and robust regression to identify the secrets.
Empirically, the approach outperforms related work.

**Strengths:**

- The work uses simple white-box models with effective pre-processing
- The main idea and modeling choices seem very interesting
- Empirically, the method outperforms related work

**Weaknesses:**

- Potentially quite preprocessing sensitive. It is hard for me to say how difficult it is to get the preprocessing right for a new problem
- Performance drop at higher dimensions
- Underperforming SOTA approaches (but being more efficient)

**Questions:**

- Can you provide ablation studies for some parts of the pipeline? So that it is easier to get a feeling of the impact of the steps
    - vector-saving
    - ...
- Could the interpretability of the model be used further? This would be a competitive advantage over the black box-related work

---

> ### Author Response · Authors · 2025-11-26
>
> We thank the reviewer for the constructive feedback and for highlighting the effectiveness of our white-box modeling approach. We are pleased that the reviewer found the main idea interesting and recognized that our method empirically outperforms related work. We address the specific concerns regarding preprocessing sensitivity and model interpretability below to clarify the robustness of our contribution.
>
> ## W1. Potentially quite preprocessing sensitive. It is hard for me to say how difficult it is to get the preprocessing right for a new problem.
>
> We thank the reviewer for this insightful observation regarding the configuration of the preprocessing pipeline. We would like to clarify that while the pipeline involves several components, it is designed to be robust and largely automated, minimizing the burden of manual tuning for new problems.
>
> The parameters can be categorized into two groups:
> - **Critical Parameters**: The success of the attack primarily depends on the penalty parameter ($\omega$) and the BKZ block size ($\beta$). Crucially, $\omega$ is derived to balance the trade-off between the secret reduction and error amplification inherent to the dual embedding, and we automatically calculate it if unset. The block size $\beta$ dictates the computational cost of the BKZ 2.0 SVP oracle; a higher $\beta$ yields better reduction (a lower root Hermite factor $\delta_0$) but at exponentially higher computational time.
> - **Optimization Parameters**: Parameters governing the vector saving strategy, progressive BKZ scheduling, and matrix embedding serve to optimize runtime efficiency and computational costs rather than being strict prerequisites for success. Most of them are quite robust for different Module-LWE parameters.
>
> In our released implementation, we provide default settings that we found to be working well across diverse LWE parameter sets (including varying dimensions and error distributions). For example, using FLATTER for initial "warm-up" rounds and setting a low reduction max size (around 50) are standard defaults that maximize efficiency. Consequently, applying the method to a new problem instance generally requires only defining the BKZ2.0 block size, with the algorithm automatically handling the internal configuration.
>
> To further facilitate this, we include a notebook, optimal\_parameters.ipynb, in the repository that allows users to verify and generate these automated parameters for any given instance.
>
> ## Q1. Can you provide ablation studies for some parts of the pipeline?
>
> We agree that isolating the contributions of individual pipeline components provides crucial insight into the attack's mechanics. In response, we have conducted three specific ablation studies to quantify the impact of our preprocessing and regression choices. We will include the resulting plots and detailed analysis in a new section in the appendix of the revised paper.
>
> The ablation studies focus on the following key aspects:
> - **Impact of Vector Saving**: We analyzed the effectiveness of the "Vector Saving Strategy" described in the paper across 11 different parameter sets. By retaining the shortest unique vectors discovered across multiple BKZ tours, we observed a consistent improvement in the quality of the output basis. Specifically, the strategy yielded an average reduction in the mean row norm of approximately 9.5\% across different settings, with a maximum of 17.6\%.
> - **MLWE Optimization**: We evaluated the trade-off inherent in our algebraic amplification techniques. The study demonstrates that these methods allow us to scale the sample pool size by a factor $n$ (dimension of each MLWE rank). Crucially, we show that this massive increase in quantity does not degrade quality: the "inlier rate" remains stable, deviating by less than 2\% on average compared to the non-amplified baseline.
> - **Robustness of Regression Algorithms**: We performed a comparative analysis of the breakdown points for different regression algorithms, specifically testing OLS, Huber, RANSAC, and Tukey’s Biweight. We found that standard RANSAC performed worse than standalone regressors in this high-dimensional setting. In contrast, Tukey’s Biweight demonstrated superior resilience, tolerating outlier rates of up to 44\% in some configurations, significantly outperforming OLS and Huber regression. This robustness incurs a higher computational cost, largely because we rely on statsmodels rather than the highly optimized scikit-learn implementations used for the other regressors. Yet, it enables our linear model to achieve outlier tolerance comparable to that of state-of-the-art transformer-based attacks like SALSA VERDE and FRESCA.

---

> ### Author Response · Authors · 2025-11-26
>
> ## Q2. Could the interpretability of the model be used further?
>
> We thank the reviewer for highlighting this aspect and agree that the "white-box" nature of linear models constitutes a distinct competitive advantage over black-box transformers. We leverage this interpretability extensively in our current pipeline: because the model weights directly correspond to the target secret vector $s$, we can enforce strong prior knowledge throughout the training process. Specifically, we initialize the weights to follow the known distribution of the secret. Furthermore, during the recovery phase, we explicitly analyze the learned weights by applying normalization, bounding, and rounding based on the specific secret distribution to effectively filter out the noise component $Re$.
>
> To address the potential for further interpretability, we investigated advanced techniques including gradient-based outlier detection — monitoring the cosine similarity of weight updates — and an unsupervised framework utilizing PCA and clustering to geometrically separate modular wrap-arounds. However, these heuristics did not yield any improvement over our robust regression baseline. We found that the robust loss function effectively mitigates outliers implicitly.

---

### Official Review · Reviewer_KqrX · 2025-11-02

**Soundness:** 2
**Presentation:** 2
**Contribution:** 2
**Rating:** 2
**Confidence:** 3

**Summary:**

This paper studies machine-learning-based attacks on the Module Learning With Errors (MLWE) problem. By exploiting the module structure, we develop several optimization techniques that improve upon existing ML-based attacks such as SALSA, SALSA PICANTE, and SALSA VERDE.

**Strengths:**

The effort to exploit the module structure as fully as possible to further optimize the attack is impressive. By leveraging structural properties of MLWE that previous ML-based attacks did not exploit, the paper amplifies the number of vectors used in the attack—an improvement that is technically meaningful. Moreover, the proposed method demonstrates superior performance on MLWE instances with small parameters compared to other ML-based attacks.

**Weaknesses:**

Because this paper addresses the security of a cryptographic scheme, showing a meaningful improvement over prior ML-based attacks alone is not sufficient to fully assess its contribution. In particular, it is necessary to determine what impact the proposed technique would have on currently deployed cryptographic parameters. However, the current manuscript does not analyze the parameters that are used in practice as standards.

Accordingly, to obtain community recognition of the proposed technique, the following analyses and estimates are required. First, the authors should state explicit assumptions for extrapolating attack costs to parameters used in real deployments of MLWE. To estimate attack runtimes for realistic (rather than toy) parameters, one needs a reasoned model of how the proposed attack’s runtime scales with input parameters. This model may be heuristic, but if so it must be accompanied by clear experimental evidence supporting the proposed scaling law. Given such a model and assumptions, the authors should provide estimated attack times for representative real-world parameters. These estimates are necessary to assess the concrete cryptanalytic impact on MLWE instantiations.

As it stands, the paper only reports experiments on toy parameters for which the attack runtime can be directly measured, so it is difficult to judge the cryptographic impact of the results. It would be helpful to refer to the following paper for guidance on how to properly analyze an attack algorithm.

Chen, Yuanmi, and Phong Q. Nguyen. "BKZ 2.0: Better lattice security estimates." International Conference on the Theory and Application of Cryptology and Information Security. Berlin, Heidelberg: Springer Berlin Heidelberg, 2011.

**Questions:**

1. Can you derive a predictive formula that estimates attack runtime for real-world (practical) parameters?
2. If so, what are the heuristic assumptions required for that extrapolation, and what experimental evidence supports those assumptions?
3. Given those assumptions, what is the estimated attack runtime for realistic parameter sets?

---

> ### Author Response · Authors · 2025-11-27
>
> We thank the reviewer for the constructive feedback and for highlighting the necessity of analyzing real-world parameters to assess the cryptographic impact of our method. We agree that establishing a link between our experimental results on small/sparse parameters and deployed cryptographic standards is essential.
>
> In response to your request, we have derived a predictive model to estimate the complexity of the *NoMod* attack on standard ML-KEM (Kyber) parameters. We have implemented this estimation within the standard [lattice-estimator](https://github.com/malb/lattice-estimator) framework and will open-source this module to facilitate future research.
>
> Below, we address your three specific questions regarding the heuristic assumptions, predictive formulas, and concrete runtime estimates.
>
> ### 1. Heuristic Assumptions and Predictive Formulas
>
> To estimate the cost, we model the attack success as a function of the *inlier rate* ($P_{in}$) — the probability that a sample, after lattice reduction, constitutes a valid linear relationship over the integers (i.e., it is not wrapped by the modulus $q$). The complexity is dominated by the BKZ reduction required to achieve a target $P_{in}$.
>
> Our extrapolation relies on two standard assumptions in lattice cryptanalysis, consistent with the estimation made for the dual hybrid attack [1]:
>
>
> - **Assumption 1 (Geometric Series Assumption)**: Given as input, a basis B of a d-dimensional lattice, BKZ is expected to follow the Gaussian Heuristic and output a vector of norm close to [2, 4]:
>     \begin{equation}
>          ||v|| \approx \delta ^ {d-1} \cdot \det(\Lambda)^{1/d}
>     \end{equation}
> - **Assumption 2 (BKZ Quality)**: BKZ with blocksize $\beta$ achieves a root Hermite factor
>     \begin{equation}
>         \delta_0  \approx \Bigl(\tfrac{\beta}{2\pi e}\,(\pi \beta)^{\frac{1}{\beta}}\Bigr)^{\frac{1}{2(\beta-1)}}
>     \end{equation}
>     This behavior is experimentally verified in previous works [3].
>
> The derivation proceeds as follows:
>
>
> - **Shortest Vector Length ($v_{min}$)**: Using the Gaussian Heuristic, the length of the shortest non-zero vector $v_{min}$ in the embedded lattice $\mathcal{L}$ of dimension $d$ and volume $\text{vol}(\mathcal{L}) = \omega^m q^{nk}$ is estimated as:
>     \begin{equation}
>         ||v_{min}|| \approx \delta_{0}(\beta)^{d} \cdot \left(\omega^{m}q^{nk}\right)^{1/d}
>     \end{equation}
>
> - **Connection to ($\sigma_{Rb}$)**: The reduction transforms the samples into a distribution with variance $\sigma_{Rb_{i}}^{2}$. By selecting the optimal penalty $\omega = \sigma_e / \sigma_s$ to balance the contributions of the error and secret, a key relationship is established:
>     \begin{equation}
>         \sigma_{Rb_{i}}^2 = \|(R_i A + qC_i)\|^2 \sigma_s^2 + \|R_i\|^2 \sigma_e^2 = \sigma_s^2 \cdot \left( \|(R_i A + qC_i)\|^2  + w^2 \|R_i\|^2 \right) = \sigma_s^2 \cdot ||v_{i}||^2
>     \end{equation}
>
> - **Inlier Rate ($P_{in}$)**: The transformed sample is modeled as a Gaussian variable $b_{i}^{\prime}\sim\mathcal{N}(0,\sigma_{Rb_{i}}^{2})$. The inlier rate $P_{in}$ is calculated using:
>     \begin{equation}
>         P_{in} = \text{erf}\left(\frac{q}{2\sqrt{2}\sigma_{Rb_{i}}}\right)
>     \end{equation}
>
> - **Cost Estimation**: Combining the Gaussian Heuristic for $||v_{min}||$ and the variance, we obtain the final formula linking the block size $\beta$ to the inlier rate:
>     \begin{equation}
>         P_{in}(\beta) \approx \text{erf}\left(\frac{q}{2\sqrt{2}\cdot\sigma_{s}\cdot \left[\delta_{0}(\beta)^{d}(\omega^{m}q^{nk})^{1/d}\right]}\right)
>     \end{equation}
>     We then search for the minimal $\beta$ that achieves the required probability and estimate the cost using the Core-SVP model: $\text{rop} \approx 2^{0.292\beta}$.
>
> We emphasize that while the formula above accurately predicts attack complexity for real-world parameters with dense vectors, it does not directly predict behavior for sparse secret vectors attacked with low block sizes (our experimental regime). For dense secrets, the attack strictly requires finding a short vector whose norm is lower than $q$ to ensure the inner product does not wrap modulo $q$; the analytical formula effectively models this by predicting when the expected $v_{min}$ drops below this threshold.
> However, our low-$\beta$ experiments operate in the pre-asymptotic regime, where the lattice basis profile retains a "Z-shape" and has not yet degraded into the smooth geometric decay assumed by the GSA. In this regime, the shortest vector in the lattice is often a trivial vector of norm $\approx q$. Consequently, we cannot analytically predict the length of the shortest non-zero (mod q) vector found by BKZ 2.0. Furthermore, in the case of sparse secrets, we might succeed even if the found vector has a norm slightly greater than $q$ (exploiting the sparsity), a nuance the standard GSA model does not capture.

---

> ### Author Response · Authors · 2025-11-27
>
> ### 2. Estimated Runtime for Real-World Parameters
>
> Using the formulas above, we integrated the *NoMod* estimation into the `lattice-estimator` [5]. The following Table compares the estimated cost (in $\log_2 \text{rop}$) of our ML-based approach against the standard Primal (uSVP) and Dual Hybrid attacks for NIST-standardized ML-KEM parameters.
>
> | Parameter Set | Primal (uSVP) | Dual Hybrid | NoMod (ML) | $\beta$ (NoMod) |
> | :--- | :---: | :---: | :---: | :---: |
> | **ML-KEM-512** | 143.8 | 139.7 | **176.9** | 526 |
> | **ML-KEM-768** | 204.9 | 196.4 | **266.7** | 846 |
> | **ML-KEM-1024** | 275.1 | 262.3 | **376.3** | 1236 |
>
> The results indicate that for standard, dense parameter sets, the complexity of *NoMod* ($\approx 177$ bits for ML-KEM-512) is strictly higher than classical lattice attacks ($\approx 140-144$ bits).
> This behavior is expected and stems from the geometric trade-offs of our embedding. While standard attacks (like Primal uSVP) target the secret or error vectors directly, our method can theoretically succeed by finding vectors of higher norm. However, the different embedding constructs a lattice with a significantly higher volume than the embeddings used in standard attacks. This volume inflation imposes a severe penalty: to compensate for the massive determinant and still find a vector within the useful norm range, the reduction algorithm must achieve a much lower root Hermite factor ($\delta_0$). Achieving this necessitates a disproportionately high block size ($\beta > 500$), thereby driving the computational complexity above that of standard methods for these dense regimes.
> Consequently, while *NoMod* is highly effective against non-standard or sparse variants, these estimates confirm that current NIST parameters remain secure against this class of machine learning attacks.
>
> ---
>
> [1] Espitau, T., Joux, A., Kharchenko, N. (2020). On a Dual/Hybrid Approach to Small Secret LWE. In: Bhargavan, K., Oswald, E., Prabhakaran, M. (eds) Progress in Cryptology – INDOCRYPT 2020. INDOCRYPT 2020. Lecture Notes in Computer Science(), vol 12578. Springer, Cham.
>
> [2] Claus-Peter Schnorr. Lattice reduction by random sampling and birthday methods. In Helmut Alt and Michel Habib, editors, STACS 2003, 20th Annual Symposium on Theoretical Aspects of Computer Science, Berlin, Germany, February 27- March 1, 2003, Proceedings, volume 2607 of Lecture Notes in Computer Science, pages 145156. Springer, 2003.
>
> [3] Yuanmi Chen. R´eduction de r´eseau et s´ecurit´e concrete du chiffrement completement homomorphe. PhD thesis, Paris 7, 2013.
>
> [4] Chen, Yuanmi, and Phong Q. Nguyen. "BKZ 2.0: Better lattice security estimates." International Conference on the Theory and Application of Cryptology and Information Security. Berlin, Heidelberg: Springer Berlin Heidelberg, 2011.
>
> [5] Martin R. Albrecht, Rachel Player and Sam Scott. On the concrete hardness of Learning with Errors.
> Journal of Mathematical Cryptology. Volume 9, Issue 3, Pages 169–203, ISSN (Online) 1862-2984,
> ISSN (Print) 1862-2976 DOI: 10.1515/jmc-2015-0016, October 2015

---

### Author Response · Authors · 2025-12-03

We would like to express our gratitude to all the reviewers for their time and insightful feedback. The review process highlighted the need to bridge the gap between our experimental success on specific parameter sets and the broader security implications for standardized cryptography. In response, we have revised the paper to include a theoretical complexity framework, targeted ablation studies, and a sharpened discussion on the distinct advantages of our white-box approach.

Below, we detail the major methodological and structural updates in the revised manuscript.

## 1. Theoretical Framework and Real-World Complexity (Section 6 \& Appendix H)

A primary objective of this revision was to rigorously quantify the impact of our attack on real-world parameters, moving beyond experimental "toy" instances. To this end, we developed and implemented the NoMod Estimator, a specialized cost model integrated into the standard lattice-estimator framework.

- **Z-Shape Simulation**: Standard estimators typically rely on the Geometric Series Assumption (GSA). However, our attack operates in the pre-asymptotic regime with low block sizes ($\beta$), where the lattice basis profile follows a "Z-shape" rather than a linear decay. We introduced a simulation framework (Appendix H) that accurately models this Z-shape profile and the variance of the reduced vectors.

- **Component Separation**: Our model explicitly separates the amplification of the error vector ($R$) from the amplification of the secret ($RA+qC$). This allows us to analytically predict the inlier rate and determine the minimum block size $\beta$ required for the robust regressor to converge.

- **ML-KEM Estimates**: In the new Section 6, we apply this estimator to NIST-standardized parameters (Table 5). The results show that for dense, standard parameters like ML-KEM-512, NoMod requires approximately $2^{164}$ rop, which is higher than the classical uSVP baseline ($\approx 2^{143}$). We explain that this increased cost stems from the volume inflation inherent in our specific lattice embedding, confirming that while NoMod is highly effective in sparse or non-standard regimes, current NIST parameters remain secure against this class of attack.

## 2. Ablation Studies (Appendix G)

To isolate the contributions of our specific engineering choices, we added Appendix G, which details three new ablation studies:

- **Vector Saving Strategy**: We quantified the impact of recycling short vectors across BKZ tours. Our experiments confirm this strategy yields a strictly lower distribution of row norms compared to standard BKZ 2.0, with an average reduction of 9.5\% and up to 17.6\% in specific settings.

- **Algebraic Amplification**: We investigated whether generating data via polynomial rotation degrades sample quality. The results (Fig. 1b) demonstrate that we can scale the sample pool by a factor of $n$ while keeping the "inlier rate" deviation below 2\%. This confirms that rotational expansion provides high-quality training data without additional lattice reduction costs.

- **Regressor Robustness**: We benchmarked OLS, Huber, RANSAC, and Tukey’s Biweight. The analysis (Fig. 2) reveals that Tukey’s Biweight is superior, tolerating outlier rates up to 44\%. This justifies its higher computational cost compared to standard OLS, as it matches the robustness of complex deep learning attacks using a purely linear model.

## 3. Reframed Conclusion (Section 7)

We have rewritten the Conclusions (Section 7) to better articulate the conceptual contributions of the work:

- **White-Box Inference**: We emphasize the paradigm shift from black-box learning to white-box inference. Unlike transformers that require statistical post-processing to interpret learned representations, our linear model establishes a direct mapping between learned weights and secret coefficients.

- **Interleaved Recovery**: We highlight a unique operational advantage of our approach: "interleaved recovery." Because training a linear regressor is computationally negligible compared to lattice reduction, we can attempt secret extraction during the preprocessing phase (e.g., after every BKZ tour). This rapid feedback loop is computationally impractical with heavy neural architectures.

- **Model-Agnostic Preprocessing**: We clarify that our preprocessing innovations (vector saving, amplification) are model-agnostic and could be used to enhance transformer-based attacks, though we prioritize linear models for efficiency and interpretability.

Finally, we updated a paragraph in Section 4 (Experimental Results) to clarify that our method was successfully tested on unconstrained (dense) CBD secrets at lower dimensions, while high-dimensional sparse benchmarks were included to ensure fair comparisons with prior work. Additionally, we released the complexity estimator logic and parameter optimization notebooks in the repository to guarantee full reproducibility.

---

### Meta-Review · Area_Chair_ezAx · 2026-01-02

**Summary:**

The paper proposed a machine-learning-based attack on the Module Learning With Errors (MLWE) problem which is the basis for many public-key crypto-systems. The proposed method combines optimized lattice preprocessing and secret key recovery using robust linear regression. Experiments were used to show that the proposed method improves upon existing ML-based attacks.

Main concerns by the reviewers:
1. Since cryptoanalysis is a quite mature area, a new meaningful attack should improve deployed cryptosytems in certain way. However, the paper does not analyze the cryptosytems that are used in practice and only compares to state-of-the-art AI attacks.
2. Potentially quite preprocessing sensitive. There is limited analysis on the tradeoffs between lattice reduction and ML estimation.
3. The method’s performance declines for denser secrets or in high-dimensional parameter settings.
4. Limited contribution to machine learning.

In summary, the paper does make progress on using ML to crypto-analysis, but not to the standard of cryptography. Combining the limited contribution to ML, I would recommend rejection.

**Reviewer Concerns:**

Main concerns by the reviewers:
1. A new meaningful attack should improve deployed cryptosytems. However, the paper does not analyze the cryptosytems that are used in practice.
2. Limited contribution to machine learning.

**Reviewer Scores:**

KqrX would increase the score, but not much.
rkL9 would keep the score.
LswJ would keep the score.
KbXn would increase the score, but not much.

---

### Decision · Program_Chairs · 2026-01-26

Reject